# WUSCHEL acts as an auxin response rheostat to maintain apical stem cells in *Arabidopsis*

Yanfei Ma[1,5], Andrej Miotk [1,5], Zoran Šutiković[1,5], Olga Ermakova[1], Christian Wenzl[1], Anna Medzihradszky[1], Christophe Gaillochet[1], Joachim Forner [1], Gözde Utan[1], Klaus Brackmann[2], Carlos S. Galván-Ampudia[3], Teva Vernoux [3], Thomas Greb [4] & Jan U. Lohmann [1]*

To maintain the balance between long-term stem cell self-renewal and differentiation, dynamic signals need to be translated into spatially precise and temporally stable gene expression states. In the apical plant stem cell system, local accumulation of the small, highly mobile phytohormone auxin triggers differentiation while at the same time, pluripotent stem cells are maintained throughout the entire life-cycle. We find that stem cells are resistant to auxin mediated differentiation, but require low levels of signaling for their maintenance. We demonstrate that the WUSCHEL transcription factor confers this behavior by rheostatically controlling the auxin signaling and response pathway. Finally, we show that WUSCHEL acts via regulation of histone acetylation at target loci, including those with functions in the auxin pathway. Our results reveal an important mechanism that allows cells to differentially translate a potent and highly dynamic developmental signal into stable cell behavior with high spatial precision and temporal robustness.

[1] Department of Stem Cell Biology, Centre for Organismal Studies, Heidelberg University, D-69120 Heidelberg, Germany. [2] Vienna Biocenter (VBC), Gregor Mendel Institute (GMI), Austrian Academy of Sciences, Dr. Bohr-Gasse 3, 1030 Vienna, Austria. [3] Laboratoire Reproduction et Développement des Plantes, University of Lyon, ENS de Lyon, UCB Lyon 1, CNRS, INRA, F-69342 Lyon, France. [4] Department of Developmental Physiology, Centre for Organismal Studies, Heidelberg University, D-69120 Heidelberg, Germany. [5]These authors contributed equally: Yanfei Ma, Andrej Miotk, Zoran Šutiković. *email: jlohmann@meristemania.org

The shoot apical meristem (SAM) is a highly dynamic and continuously active stem cell system responsible for the generation of all above ground tissues of plants. The stem cells are located in the central zone and are maintained by a feedback loop consisting of the stem cell promoting WUSCHEL (WUS) homeodomain transcription factor and the restrictive CLAVATA (CLV) pathway[1,2]. WUS protein is produced by a group of niche cells, called organizing center, localized in the deeper tissue layers of the meristem[3] and moves to stem cells via plasmodesmata[4,5]. WUS is required for maintaining stem cells and SAMs of *wus* mutants terminate due to stem cell exhaustion after producing a small number of organs[6]. Conversely, mutants in genes of the *CLV* pathway exhibit substantial stem cell over-proliferation, which is strictly dependent on *WUS* activity[1,2]. *CLV3* is the only component of this system that is specifically expressed in stem cells and hence serves as a faithful molecular marker. Stem cells are surrounded by transient amplifying cells, which are competent to undergo differentiation in response to auxin, a small, mobile signaling molecule with diverse and context specific roles in plant development and physiology (reviewed in ref. [7]). Auxin sensing is dependent on nuclear receptors including *TRANSPORT INHIBITOR RESPONSE1 (TIR1)*, whose activation triggers the proteolytic degradation of AUX/IAA proteins, such as BODENLOS (BDL). AUX/IAA proteins repress auxin responses by inhibiting the function of activating AUXIN RESPONSE FACTOR (ARF) transcription factors via dimerization[8–10]. Intracellular accumulation of auxin is regulated by active polar transport and in the context of the SAM, the export carrier PINFORMED1 (PIN1) determines the sites of lateral organ initiation and thus differentiation[11,12]. In addition to promoting organ initiation, auxin influences stem cell proliferation by interacting with the signaling cascade of another classical phytohormone, cytokinin, and allows lateral organs to communicate with the center of the meristem[13–15].

Here, we ask how long-term stem cell fate is robustly maintained within a tissue environment that is subject to such a highly dynamic signaling system geared towards differentiation. We find that stem cells are resistant to auxin mediated differentiation, but require low levels of signaling for their maintenance. Using genomic and genetic approaches, we show that the WUSCHEL transcription factor confers this behavior by rheostatically controlling the auxin signaling and response pathway. Finally, we demonstrate that WUSCHEL acts via regulation of histone acetylation at target loci, including those with functions in the auxin pathway.

## Results

**Role of auxin signaling for apical stem cell fate.** To analyze auxin distribution and response with cellular resolution across the homeostatic apical stem cell system of *Arabidopsis*, we mapped auxin signaling behavior using the genetically encoded markers R2D2 and DR5v2 (ref. [16]). R2D2 is based on a fusion of the auxin-dependent degradation domain II of an Aux/IAA protein to Venus fluorescent protein, and uses a mutated, non-degradable domain II linked to tdTomato as an internal control[16]. Hence, R2D2 signal is dictated by the levels of auxin, as well as the endogenous receptors and represents a proxy for the auxin signaling input for every cell. Following multispectral live-cell image acquisition in plants carrying R2D2, we used computational analysis of the yellow to red ratio to determine the cellular auxin input status. We found that auxin is present and sensed fairly uniformly across the SAM including the central stem cell domain, with local minima only detected at young primordia and developing organ boundaries (Fig. 1a, b and refs. [17,18]). In contrast, DR5v2, a reporter for auxin signaling output based on a synthetic promoter containing repeats of ARF DNA binding motifs, was strongly activated non-uniformly in wedge shaped zones of differentiation competent cells, but only weakly expressed in the center of the SAM (Fig. 1d; ref. [17]). To spatially correlate cellular auxin output status with stem cell identity, we combined the DR5v2 reporter with a *pCLV3:mCherry-NLS* marker in a single transgenic line. Computational analysis of the DR5v2 and *pCLV3* signals revealed that the auxin response minimum invariantly coincided with the center of the stem cell domain (Fig. 1c–f).

To test if the auxin output minimum is functionally connected to stem cell identity, we interfered with their maintenance. To this

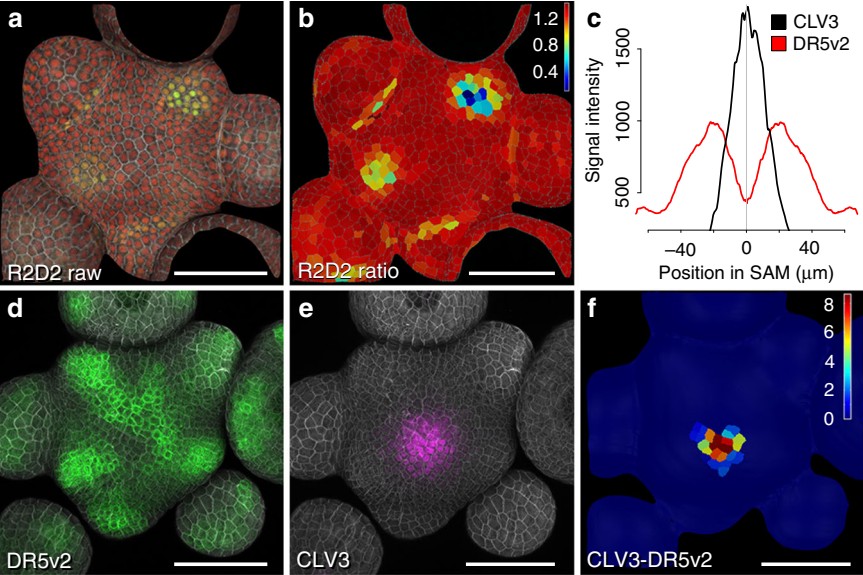

**Fig. 1** Auxin output minimum correlates with apical stem cells. **a** Confocal readout from R2D2 auxin input sensor. **b** Ratiometric representation of R2D2 activity in the epidermal cell layer (L1). **c** Quantification of averaged *pDR5v2:ER-eYFP-HDEL* and *pCLV3:mCherry-NLS* distribution (*n* = 5). **d** Confocal readout from *pDR5v2:ER-eYFP-HDEL* auxin output reporter. **e** *pCLV3:mCherry-NLS* stem cell marker in the same SAM. **f** Computational subtraction of L1 signals shown in **d** and **e**. Relative signal intensity is shown in arbitrary units. Scale bars: 50 μm

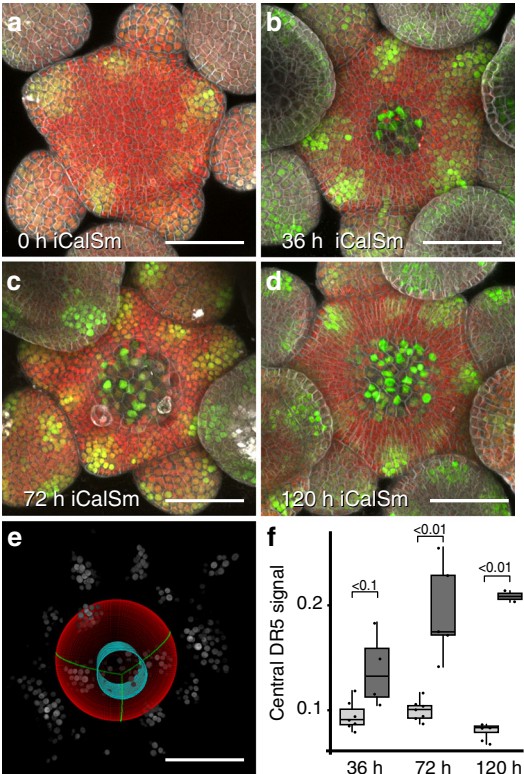

**Fig. 2** The central auxin signaling minimum is dependent on stem cell fate. **a–d** *pDR5v2:3xVENUS-NLS* activity after stem cell specific induction of iCalSm. Stem cell differentiation is marked by loss of *pRPS5a:NLS-tdTomato*. Independent channels and Z-projections are shown in Supplementary Fig. 1. **e** Computational sphere fitting and identification of the central zone for fluorescence signal quantification (see Supplementary Fig. 2). **f** Quantification of DR5v2 signal intensity in the central zone across the experimental cohort described in Supplementary Fig. 2. Light gray boxes represent uninduced controls, dark gray boxes represent plants induced with 1% ethanol. Individual analyzed SAMs are indicated as dots. Analysis of variance (ANOVA) showed significant variation between control and induced samples. Post hoc Turkey test *p* values are shown. Scale bars: 50 μm. See also Supplementary Fig. 2

end, we experimentally induced symplastic isolation through callose deposition at plasmodesmata of stem cells[19]. This treatment leads to stem cell differentiation due to restriction of WUS cell-to-cell mobility within hours after onset of callose synthase expression[5,19]. Following DR5v2 signal over time, we observed activation of auxin signaling output in the central zone domain after 36 h of callose synthase (iCalSm) expression from the *CLV3* promoter. In addition, cell expansion, a hallmark of plant cell differentiation, became obvious after 72 h (Fig. 2a–d, Supplementary Fig. 1). All plants that exhibited stem cell loss following iCalSm activation showed this pattern, which also led to a significant increase in central DR5v2 signal intensity over time, in contrast to controls that did not respond (Fig. 2e, f; Supplementary Fig. 2).

Thus, stem cell fate and the auxin response minimum appeared to be functionally connected, leading us to hypothesize that manipulation of auxin signaling in the central zone should affect stem cell behavior. To test this directly, we designed a transgene to suppress auxin signaling output specifically in stem cells. Therefore, we fused the dominant auxin signaling output inhibitor *BDL-D* (IAA12)[20] with the glucocorticoid receptor tag. The activity of the resulting fusion protein could be induced by dexamethasone (DEX) treatment, which allowed the

translocation of BDL-D-GR from the cytoplasm to the nucleus, its native cellular compartment[21]. In line with our expectations, we found that inducing *pCLV3:BDL-D-GR* led to an expansion of the DR5v2 minimum in the center of the SAM reflecting the inhibitory activity of BDL-D on ARF transcription factors (Fig. 3a, b). Surprisingly, long term induction of BDL-D-GR or stem cell specific expression of *BDL-D* without the *GR* tag caused meristem termination (in 45 of 90 independent *pCLV3:BDL-D* T1 plants; Fig. 3c, d), demonstrating that stem cells require active auxin signaling for their maintenance. Since the forced expression of transcriptional regulators, such as BDL-D, may not only interfere with auxin signaling output, but may also cause a switch in cell fate independently of signaling, we tested in which temporal order auxin output and stem cell fate were affected by BDL-D. To this end we analyzed the effect of BDL-D induction on DR5v2 and *pCLV3:mCherry-NLS* activity by time resolved live cell imaging. Whereas DR5v2 signal was clearly reduced already 24 h post induction in some individuals, activity of the *CLV3* stem cell marker was enhanced up to 72 h and then started to fade (Fig. 3e–l). Meristem size was reduced starting around the 120 h timepoint and *CLV3* signal was lost in some SAMs that appeared to be terminating at 168 h. This experiment demonstrated that the loss of auxin signaling output clearly precedes an elevation of *CLV3* expression and allowed us to rule out that the expression of BDL-D forced stem cells into differentiation independently from its role in auxin signaling.

Having established that stem cells require auxin signaling for their activity, we next tested the response of elevating auxin output. To our surprise, expression of a potent positive signaling component, the auxin response factor *ARF5/MONOPTEROS* (*MP*), or its constitutively active form *MPΔ*, which engages the auxin pathway independently of signal perception[22,23], did not cause relevant reduction in meristem size (*n* = 120, Fig. 3m–p and ref. [15]). However, when expressed throughout the entire SAM by the HMG promoter (Supplementary Fig. 3a, b), *MPΔ* stimulated ectopic organ initiation specifically in the peripheral zone (in 38 of 80 independent *pHMG:MPΔ* T1 plants; Fig. 3s, t), demonstrating that resistance to auxin was not a general feature of the meristem, but limited to stem cells. Importantly, the DR5v2 reporter, which senses auxin output by providing binding sites for ARF transcription factors, was activated in stem cells of plants expressing *MP* (in 6 of 12 independent T1 lines) and *MPΔ* (in 6 of 8 independent T1 lines) (Fig. 3m–o, q, r and Supplementary Fig. 3c–k), confirming the activity of our transgenes and suggesting that the resistance to auxin occurs, at least in part, downstream of ARF activity.

Taken together, these experiments demonstrated that auxin signaling is locally gated to permit a low instructive output level, while at the same time protecting stem cells from the differentiation inducing effects of the phytohormone at high signaling levels.

**WUSCHEL controls auxin signaling output in stem cells.** Since suppressing auxin signaling output in stem cell caused SAM arrest and a phenotype highly similar to *wus* mutants (Fig. 3c, d), we tested the contribution of *WUS* to controlling auxin responses in diverse genetic backgrounds. The *WUS* expression domain is massively enlarged in *clv* mutants[1,2], which causes stem cell over-proliferation phenotypes, and therefore SAMs from these plants provide an ideal background to elucidate the functional connection of WUS and auxin. Consequently, we analyzed auxin output in *clv3* meristems and found the DR5v2 minimum expanded in line with the overaccumulation of WUS, however some weak signal remained throughout the SAM (Fig. 4a, b). To test whether auxin signaling is required for stem cell over-proliferation in *clv3*

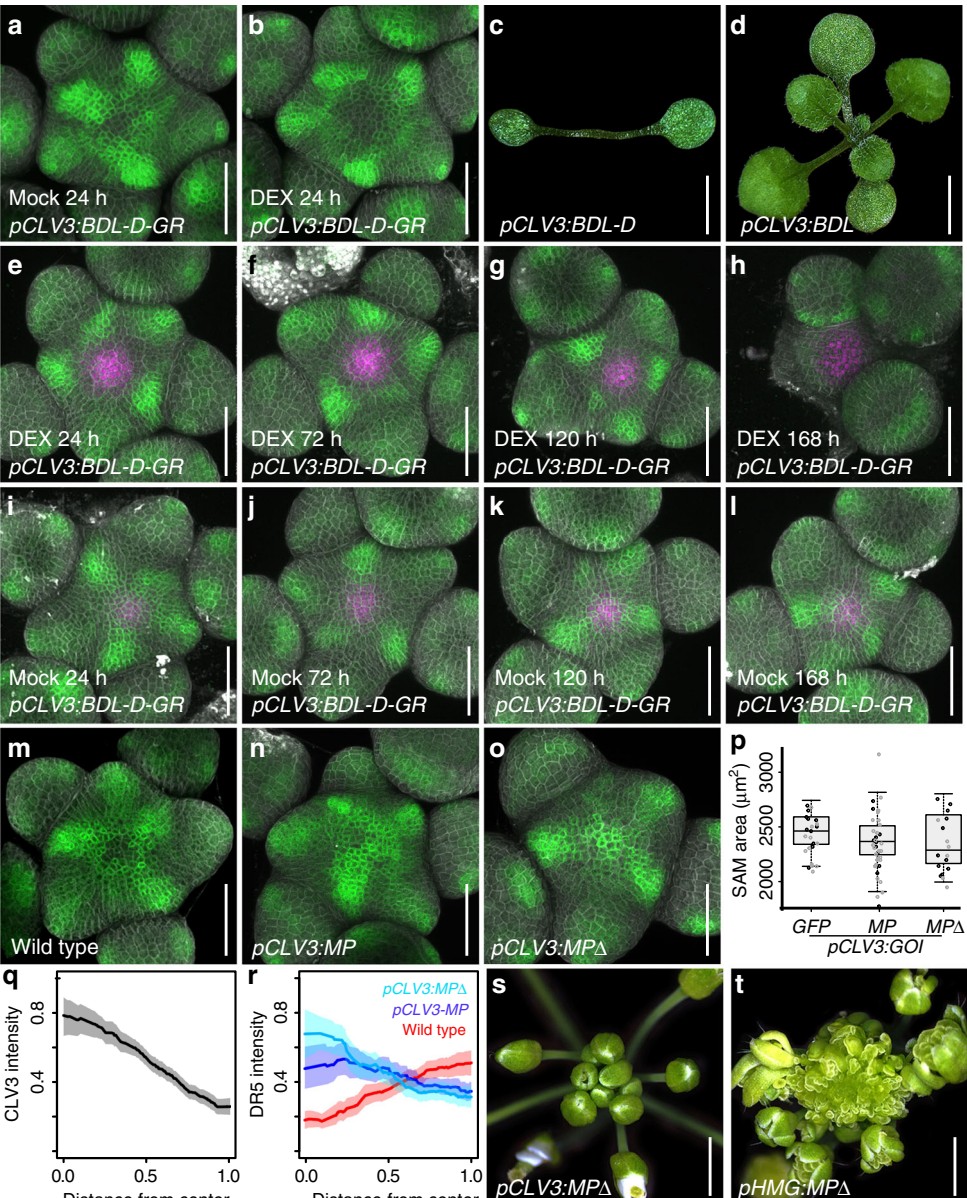

**Fig. 3** Stem cells require auxin signaling to remain active, but are resistant to overactivation of the pathway. **a** *pDR5v2:ER-eYFP-HDEL* activity in plants harboring *pCLV3:BDL-D-GR* after 24 h of mock treatment. **b** DR5v2 signal after 24 h of DEX treatment. **c, d** Representative phenotypes of lines expressing *pCLV3:BDL-D* (**c**) and *pCLV3:BDL* (**d**). **e–l** Response of DR5v2 and CLV3 to induction of *pCLV3:BDL-D-GR* over time. *pDR5v2:ER-eYFP-HDEL* shown in green, *pCLV3:mCherry-NLS* shown in magenta. CLV3 signal is expanded and sustained until the meristem terminates. **i–l** Mock treated *pCLV3:BDL-D-GR* controls. **m–o** Activity of DR5v2 in SAMs with enhanced auxin signaling in the central zone. **m** wild type control, (**n**) *pCLV3:MP*, (**o**) *pCLV3:MPΔ*. **p** SAM size quantifications for plants carrying *pCLV3:GFP*, *pCLV3:MP*, or *pCLV3:MPΔ* in two independent T1 populations. **q** Quantification of *pCLV3:mCherry-NLS* signal strength in wild type (*n* = 4). **r** Quantification of *pDR5v2:ER-eYFP-HDEL* signal strength in wild type (red, *n* = 4), *pCLV3:MP* (light blue, *n* = 4) and *pCLV3:MPΔ* (purple, *n* = 4). In **q** and **r** kernel regression was used to visualize dependence of fluorescence signal from SAM center. 95% confidence intervals were simulated using bootstrapping (10.000 iterations) and are shown in lighter colors. **s, t** Representative phenotypes of lines expressing *pCLV3:MPΔ* (**s**) and *pHMG:MPΔ* (**t**). All scale bars 50 μm, except **c** and **d** 2 mm; (**s**) and (**t**) 3.5 mm

mutants, we locally blocked auxin output by our *pCLV3:BDL-D* transgene and observed stem cell termination phenotypes in all plants (*n* = 56). While in most individuals, SAMs arrested already during the seedling stage, some developed an inflorescence before termination (*n* = 3, Fig. 4c). This result suggested that even in fasciated SAMs of *clv3* mutants, ectopic WUS is sufficient to reduce auxin signaling, while at the same time permitting basal output levels. To test the short-term effect of enhancing WUS levels without the indirect effects of the *clv3* phenotype, we created plants that carry a *pUBQ10:mCherry-GR-linker-WUS (WUS-*

*GR)* transgene which allowed for experimental induction of ubiquitous WUS activity. Long term inductions provoked phenotypes described for transgenes expressing the *WUS* cDNA without tag and included an enlarged SAM consisting of more cells after 4 days (Supplementary Fig. 4). After 24 h of DEX treatment the central auxin signaling minimum, as well as the *CLV3* domain expanded (Fig. 4d–f; Supplementary Fig. 5a–f), suggesting that WUS is indeed sufficient to reduce signaling output in the center of the SAM, but is unable to override active auxin responses at the periphery. To test whether *WUS* is also

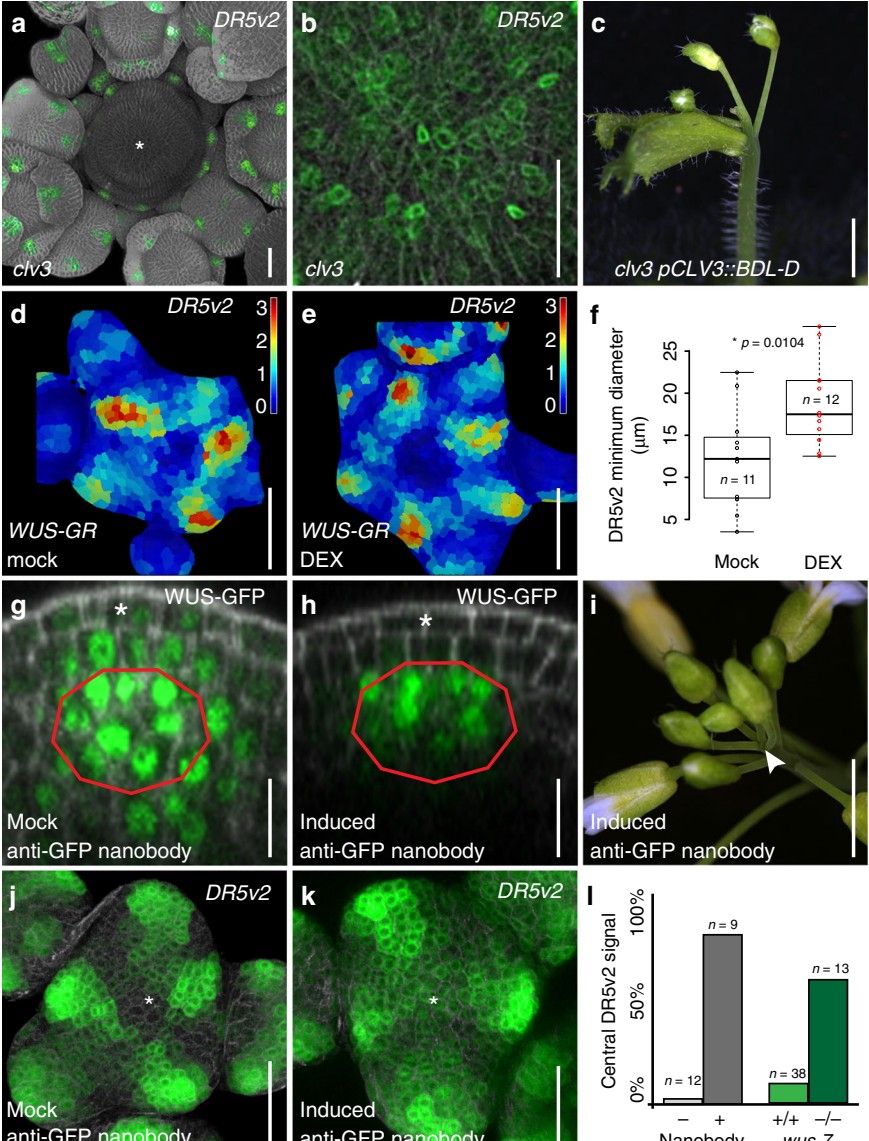

**Fig. 4 WUSCHEL maintains low auxin signaling output in stem cells. a** *pDR5v2:ER-mCherry-HDEL* activity in SAM of *clv3* mutant. Asterisk marks center of SAM. **b** Zoom into central SAM area of *clv3* mutants reveals basal *pDR5v2* activity. **c** SAM arrest caused by *pCLV3:BDL-D* expression in *clv3*. **d, e** Representative *pDR5v2:ER-eYFP-HDEL* signals after 24 h of mock treatment (**d**) or inducible ectopic activation of *WUS-GR* activity (**e**). **f** Quantification of central DR5v2 signal minimum size in µm following ectopic WUS activation. *p*-value derived by *t*-test. **g, h** Representative images of a *pWUS:WUS-linker-GFP* rescue line expressing the anti GFP nanobody under the control of *pCLV3:AlcR* (*wus/pWUS:WUS-linker-GFP/pCLV3:AlcR/pAlcA:NSlmb-vhhGFP4*). **g** WUS-linker-GFP signal after 24 h of mock treatment. **h** WUS-linker-GFP signal after 24 h of WUS depletion. Red lines mark *WUS* mRNA expressing cells of the organizing center; asterisk denote epidermal stem cells. **i** Shoot termination observed five days after WUS depletion. **j, k** Representative *pDR5v2:ER-eYFP-HDEL* signals after 24 h of mock treatment (**j**) or depletion of WUS protein from stem cells (**k**). **l** Quantification of DR5v2 presence in the central zone following WUS depletion or in weak *wus-7* mutants. All scale bars 50 µm except **c** and **i** 1 cm; **g** and **h** 10 µm

required to protect stem cells from high signaling levels, which lead to differentiation, we developed a genetic system that allowed us to inducibly degrade WUS protein in stem cells. To this end, we adapted deGradFP technology[24] and combined switchable stem cell specific expression of an anti-GFP nanobody with a *pWUS:WUS-linker-GFP wus* rescue line[5]. After 24 h induction of nanobody expression, WUS-linker-GFP signal was substantially reduced in stem cells of the epidermis and subepidermis (Fig. 4g–h) and after five days we observed shoot termination (Fig. 4i). Combining this *wus/pWUS:WUS-linker-GFP/pCLV3: AlcR/pAlcA:NSlmb-vhhGFP4* line with the DR5v2 marker showed that after 24 h of WUS depletion, cells in center of the SAM had become responsive to auxin whereas they remained insensitive in

mock treated controls (Fig. 4j–l). We made similar observations in plants carrying DR5v2 and the weak *wus-7* allele, which were able to maintain a functional SAM for some time and only terminated stochastically. In these lines, DR5v2 activity fluctuated substantially and was frequently observed in the central zone (Fig. 4l and Supplementary Fig. 6). Taken together, these results demonstrated that WUS is required to rheostatically maintain stem cells in a state of low auxin signaling.

**Mechanisms of auxin pathway gating**. To address how WUS is able to control the output of the auxin pathway, we went on to define the repertoire of direct target genes combining new

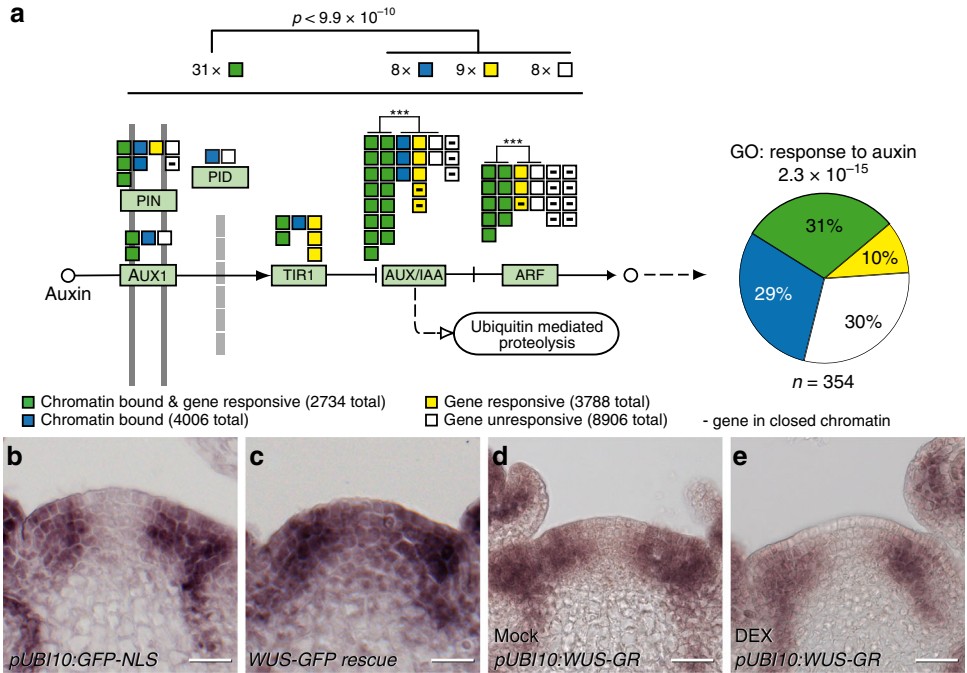

**Fig. 5** Pathway level control underlies WUSCHEL mediated gating of auxin signaling. **a** RNA-seq and ChIP-seq data demonstrate that WUS globally affects the auxin pathway, including transport, perception, signal transduction, as well as transcriptional response. Genes involved in the auxin signaling pathway are depicted as color-coded boxes, data on WUS binding and transcriptional response are shown in Supplementary Table 2. Across the entire pathway, WUS bound and responsive genes are overrepresented ($p < 9.9 \times 10^{-10}$ Fisher exact test for $2 \times 2$ matrix, bound and responsive genes (green color code) vs. all other classes, open chromatin as a background). Even within gene families, such as *AUX/IAA* or *ARF*, WUS targets are overrepresented (***$p < 10^{-4}$ by Fisher exact test for $2 \times 2$ matrix, bound and responsive genes (green color code) vs. all other classes, open chromatin as a background). Genes downstream this pathway are annotated as "responsive to auxin" and are also highly overrepresented among WUS targets (GO term enrichment taken from Supplementary Table 1). **b**, **c** *MP* RNA accumulation 24 h post anti-GFP nanobody induction in a *pUBQ10:GFP-NLS* control line (**b**) and the *pWUS:WUS-linker-GFP wus* rescue background (**c**). **d**, **e** Response of *MP* mRNA to ectopic activation of WUS-GR. *MP* RNA after 24 h of mock (**d**) or DEX treatment (**e**). Scale bars 20 μm

ChIP-seq and RNA-seq experiments using seedlings of our *WUS-GR* line. Leveraging the uniform and moderate expression transgene, the tight inducibility afforded by the linker-GR fusion protein, as well as the high affinity of RFP-trap single chain antibodies to the mCherry tag used for our ChIP protocol, we were able to identify 6740 genomic regions bound by WUS. This compared to 136 regions we had previously identified by ChIP-chip[25]. Previously identified direct targets, such as *ARR7, CLV1, KAN1, KAN2 AS2*, and *YAB3* (refs. [25–27]) were also picked up in our new datasets. Interestingly, WUS binding was almost exclusively found in regions of open chromatin[28] and among the WUS targets we found the gene ontology term "response to auxin" to be most highly enriched within the developmental category (Supplementary Table 1). Importantly, WUS appeared to control auxin signaling output at all relevant levels, since it was able to bind to the promoters or regulate the expression of a large number of genes involved in auxin transport, auxin perception, auxin signal transduction, as well as auxin response, which occurs downstream of ARF transcription factors (Fig. 5a; Supplementary Tables 2 and 3). Since WUS can act as transcriptional activator or repressor dependent on the regulatory environment[29,30] and our profiling results were based on ectopic expression of WUS in non-stem cells, we were unable to predict how the expression of individual targets would be affected in vivo. However, it has been reported that in the SAM, WUS mainly acts as a transcriptional repressor[25–27,29] and consistently, many auxin signaling components are expressed at high levels only in the periphery of the SAM and exhibit low RNA accumulation in the cells that are positive for WUS protein[17]. To test if WUS is required for this pattern, we analyzed the response of *MP* and *TIR1* mRNA

accumulation to variations in *WUS* expression. To circumvent morphological defects of stable *wus* mutants, we again made use of our deGradFP line to analyze expression of *MP* after loss of WUS protein activity, but prior to changes in SAM morphology. After 24 h of WUS depletion, *MP* mRNA expression had extended from the periphery into the central zone (Fig. 5b, c; Supplementary Fig. 7), demonstrating that WUS is indeed required for *MP* repression in stem cells. Conversely, ectopic activation of WUS revealed that it is also sufficient to reduce, but not shut down *MP* and *TIR1* transcription even in the periphery of the SAM (Fig. 5d, e, Supplementary Fig. 5g–j).

To elucidate the molecular mechanisms responsible for the observed rheostatic activity, we asked whether chromatin structure may be changed in response to WUS. WUS physically interacts with TOPLESS (TPL)[31,32], a member of the GROU-CHO/Tup1 family of transcriptional co-repressors. These adaptor proteins mediate interaction with HISTONE DEACETYLASES (HDACs, reviewed in ref. [33]), which in turn act to reduce transcriptional activity of chromatin regions via promoting the removal of acetyl modifications from histone tails[34]. To test whether regulation of chromatin modification is involved in translating WUS activity into the observed reduction of transcriptional activity of target genes we quantified histone acetylation on H3K9/K14 and methylation on H3K27. After 2 h of induction of our *WUS-GR* line, we observed a significant change in the genome wide histone acetylation patterns, which were spatially correlated with WUS chromatin binding events (2939 out of 6740 WUS bound chromatin regions showed acetylation changes), while histone methylation patterns were largely unaffected (634 out of 6740 WUS bound chromatin

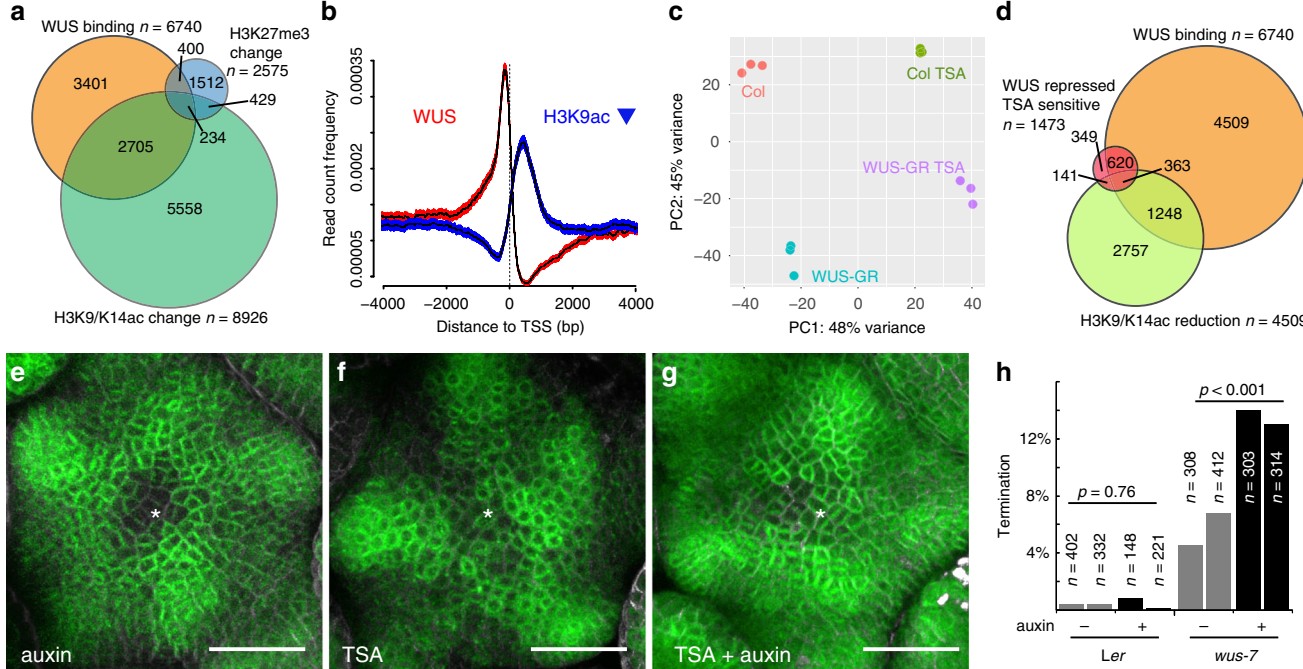

**Fig. 6 WUS acts by regulating the histone acetylation status of target loci. a** Venn diagram showing the overlap between WUS binding regions (orange), and loci with significant changes in H3K9K14ac (green) or H3K27me3 (blue) status. **b** Spatial correlation between WUS chromatin binding events (red) and regions with reduced histone acetylation (blue) 0.95 confidence intervals are shown. **c** PCA showing the global transcriptional response to WUS-GR activation in the presence or absence of TSA. TSA treatment suppressed almost 50% of gene expression variance caused by activation of WUS-GR. **d** Venn diagram showing the overlap between WUS binding regions (orange), and loci with significant reduction in H3K9K14ac (light green) and genes whose expression was reduced by WUS in a TSA sensitive manner (red). **e–g** Representative images of *pDR5v2:ER-eYFP-HDEL* activity in response to HDAC inhibition. **e** Auxin treated SAM; **f** TSA treated SAM; **g** TSA and auxin treated SAM. Asterisk denote center of the SAM. **h** Quantification of terminated seedlings grown on auxin plates from two independent experiments. On mock 5.8% seedlings ($n = 720$) segregating *wus-7* terminated; on 10 μM IAA 13.3% seedlings ($n = 617$) segregating *wus-7* terminated. Genotyping revealed that all arrested plants were homozygous for *wus-7*. For L*er* controls 0,4% seedlings arrested on mock ($n = 734$) and 0.5% on 10 μM IAA ($n = 369$). *p*-values calculated by chi-square test. Scale bars 30 μm

regions showed methylation changes) (Fig. 6a). WUS binding events clustered in the proximal promoter regions, while chromatin regions whose acetylation levels were changed after WUS activation were mainly found around the transcriptional start sites and 5′UTRs of genes (Fig. 6b). Zooming in on the 1684 directly repressed WUS targets, we found that 596 of them also showed histone de-acetylation. For the vast majority of these loci the observed reduction was fairly subtle, suggesting that mild de-acetylation may be the mechanism that allows WUS to reduce, but not shut off transcription of target genes. To test whether the observed changes in chromatin state of direct WUS targets also translate to variation in gene expression, we induced WUS activity in the absence or presence of Trichostatin A (TSA), a potent inhibitor of class I and II HDACs[35], and recorded the transcriptional response. Principle Component Analysis (PCA) not only showed that both WUS activation and TSA contributed to gene expression variance, but that there was a clear interaction of their activities. Strikingly, roughly 40% of gene expression variance caused by WUS activation was suppressed by TSA treatment (Fig. 6c). Consistently, from the 1684 directly repressed genes, 983 were no longer responsive to WUS-GR induction when TSA was present and roughly a third of them showed significant reduction in H3K9/K14 acetylation levels (Fig. 6d). These results underlined the relevance of histone de-acetylation for the genome-wide functional output of WUS and prompted us to investigate whether this mechanism is relevant for controlling auxin responses in the SAM. Therefore, we analyzed DR5v2 reporter activity after TSA and/or auxin treatment and found that auxin was insufficient to trigger a transcriptional response in stem

cells, likely due to the presence of functional WUS (Fig. 6e). In contrast, inactivation of HDACs and consequently WUS-mediated transcriptional repression by TSA treatment, led to low but consistent DR5v2 signal in the center of the meristem (Fig. 6f). Finally, combining a reduction in WUS function by TSA with stimulation of the auxin pathway caused a substantial DR5v2 response in stem cells (Fig. 6g). Taken together, these results showed that WUS binds to and reduces transcription of the majority of genes involved in auxin signaling and response via de-acetylation of histones and thus is able to rheostatically maintain pathway activity in stem cells at a basal level.

**Pathway wide control provides robustness to apical stem cell fate.** We next wondered what the functional relevance of the observed pathway wide regulatory interaction might be. Therefore, we tested the capacity of WUS targets with auxin signaling or response functions to interfere with stem cell activity. Based on their highly localized expression at the periphery of the SAM[17], we selected the signaling components *ARF3*, *ARF4*, *ARF5* (*MP*), *IAA8*, *IAA9*, and *IAA12* (*BDL*), as well as the TIR1 receptor along with transcription factors of the auxin response category including *TARGET OF MONOPTEROS* (*TMO*) and *LATERAL ORGAN BOUNDARIES* (*LOB*) genes that have established roles in other developmental contexts[36]. Neither of the 17 factors tested caused meristem phenotypes when expressed in stem cells (Fig. 3 and Table 1), highlighting the robustness of stem cell fate in the presence of WUS on the one hand and the activity of auxin signaling in these cells on the other hand. This conclusion is

**Table 1 WUS targets functionally tested by stem cell specific expression**

| AGI | Name | Responsive to auxin | Expression PZ > CZ | Promoter bound by WUS | Responsive to WUS |
|------|------|---------------------|--------------------|-----------------------|-------------------|
| AT3G62980 | TIR1 | x | x | x | x |
| AT2G33860 | ARF3 | x | x | x | x |
| AT5G60450 | ARF4 | x | x | x | x |
| AT1G19850 | ARF5 (MP) | x | x | x | x |
| AT2G22670 | IAA8 | x | x | – | x |
| AT5G65670 | IAA9 | x | x | x | x |
| AT1G04550 | IAA12 (BDL) | x | x | – | – |
| AT5G60200 | TMO6 | x | x | x | x |
| AT1G74500 | TMO7 | x | – | – | – |
| AT3G25710 | TMO5 | x | – | – | x |
| AT4G23750 | TMO3 | x | – | x | x |
| AT1G68510 | LBD42 | – | – | x | – |
| AT3G49940 | LBD38 | – | – | x | x |
| AT3G58190 | LBD29 | x | – | – | – |
| AT3G11280 | | x | x | x | x |
| AT3G28910 | MYB30 | x | x | x | x |
| AT5G58900 | | x | x | x | – |

Expression domains in the SAM are based on refs. [17,62,63]. Neither of the genes caused visible phenotypes when expressed from the *CLV3* promoter. Representative SAMs are shown in Fig. 3d, j, k, l, o

based on two observations: (1) The auxin sensitive native version of BDL was unable to terminate the SAM in contrast to the auxin insensitive BDL-D version (Fig. 3c, d). (2) *pCLV3:MP* plants showed enhanced DR5v2 activity in stem cells (Fig. 3i, j) demonstrating that ARF activity is indeed limiting for transcriptional output in wild-type. However, the transcriptional output registered by the DR5v2 reporter was not translated into an auxin response, since WUS limited the expression of a large fraction of the required downstream genes (Fig. 5a; Supplementary Table 2). Thus, WUS seems to act both upstream and downstream of the key ARF transcription factors.

Since we had found that stem cell specific expression of individual auxin signaling components was not sufficient to interfere with stem cell fate, we wanted to test whether reducing *WUS* function would sensitize stem cells to activation of the entire pathway. To this end, we grew plants segregating for *wus-7* on plates supplemented with auxin. Eleven days after germination, we observed twice as many terminated *wus-7* mutant seedlings on plates containing 10 μM IAA compared to control plates, whereas wild-type seedlings were unaffected even at 20 μM IAA (Fig. 6h, Supplementary Fig. 8). Thus, reducing *WUS* function allowed activation of auxin responses under conditions that were tolerated in wild type. Taken together, the activation of individual pathway components was insufficient to override the protective effect of WUS, however compromising the master regulator itself rendered stem cells vulnerable to even mild perturbations in auxin signaling.

## Discussion

Several previous studies have connected WOX gene activity with auxin signaling in shoot and root stem cell maintenance[13,15,37,38]. Our results now show that WUS restricts auxin signaling in apical stem cells by pathway-wide transcriptional control, while at the same time allowing instructive low levels of signaling output. This rheostatic activity may be based on selective transcriptional repression/activation of a subset of signaling and response components that render the pathway unresponsive to high input levels. Alternatively, WUS may be able to reduce expression of targets rather than to shut off their activity completely, leaving sufficient capacity for low level signaling only. In support of the latter hypothesis, we demonstrate that WUS acts via de-acetylation of histones and that interfering with HDAC activity

triggers auxin responses in stem cells. This activity would be counteracted by ARF transcription factors including MP, which has been shown to act via promoting chromatin acetylation to induce flower primordium fate at the periphery of the SAM[39]. Since there is evidence supporting both scenarios for WUS function[25,27,29,30] it appears likely that both mechanisms work hand in hand dependent on the regulatory environment of the individual cell. Thus, a definitive answer will require inducible WUS loss of function approaches in stem cells coupled with time-resolved whole genome transcript profiling at the single cell level. In contrast to our data, previous work had shown that following suppression of *WUS* activity SAMs terminate without detectable increase in auxin signaling output in the central zone[40]. This discrepancy is likely due to the use of different versions of the DR5 reporter, since the DR5v2, which has binding sites optimized for interaction with MP, is substantially more sensitive when compared to the original DR5[16]. In addition, we found that targeting the fluorescent protein to the ER enhances the detection of weak signals, such as the one in the central zone (Supplementary Fig. 3).

In addition to its effects on auxin signaling, WUS enhances cytokinin responses via the repression of negative feedback regulators[26]. This interaction can be overridden by expression of constitutively active versions of these negative feedback components[26], and similarly we find here that dominant negative auxin regulators lead to SAM arrest. In contrast, wild-type or constitutively active auxin signaling elements do not lead to SAM defects, suggesting that WUS acts primarily to limit auxin responses. Thus, by acting on both pathways by direct reduction of target gene expression, WUS protects stem cells from auxin mediated differentiation, while at the same time enhancing cytokinin output, which may primarily serve to sustain *WUS* expression[41,42]. Auxin and cytokinin signaling are directly coupled also in other stem cell systems and balancing their outputs is key to maintaining functional plant stem cell niches[15,43]. Along these lines, auxin may also be required to stabilize the WUS-CLV3 feedback loop. This idea is supported by our findings that following local suppression of auxin signaling, stem cell fate was able to expand into the peripheral zone. The resulting increase in *CLV3* levels may have caused meristem termination via reduction of *WUS* expression[1,2]. Along these lines, Luo and colleagues have shown that MP acts on *CLV3* expression via the DORN-ROESCHEN transcription factor[13]. However, since a reduction of

auxin signaling output also leads to stem cell termination in plants lacking *CLV3*, the regulatory landscape might be even more complex. Taken together and given the dynamic and self-organizing nature of auxin signaling on the one hand and the *WUS-CLV* feedback loop on the other hand[1,2,44], the close interaction between these two systems appears to be required to provide long-term robustness to the apical stem cell niche of plants. In this scenario, WUS would act as the central integrator, owing to its rheostatic influence on target gene activity.

## Methods

**Plant material and treatments**. All plants were grown at 23 °C in long days or continuous light. Ethanol inductions were performed by watering with 1% ethanol and continuous exposure to ethanol vapor, refreshed every 12 h. WUS-GR was induced by submerging seedlings in 10 μM dexamethasone, 0.015% Silwet L-70 in 0.5× MS for 2 h. For local induction of WUS-GR or BDL-D-GR at the SAM, 10 μl induction solution were directly applied to the primary inflorescence meristem. Auxin plates were 0.5× MS, 1% agar, pH 5.7, 10 μm IAA. For TSA/IAA cotreatments, shoot apical meristems were dissected from about 4 cm high stem and cultured in vitro in Apex Growth Medium (AGM) overnight[45]. AGM was supplemented with vitamins (Duchefa M0409), cytokinin (200 nM 6-Benzylaminopurine), and IAA (3-indole acetic acid, 1 mM) and/or Trichostatin A (TSA, Sigma, T8552, final concentration 5 μM) or mock before imaging. IAA stock solution (0.1 M in 0.2 M KOH) was diluted with 2 mM M.E.S (pH 5.8) to 1 mM working solution, then added to the plates for 30 min before imaging on the second day.

For WUS-induction with TSA treatments, seedlings were submerged in DEX (10 μM) or TSA (1 μM) solution or both, slowly shaken for 2 h, and then harvested for RNA-seq.

All plants were of Col-0 accession apart from *wus-7*, which was in L*er* background. For experiments involving *wus-7*, L*er* plants were used as controls. *CLV3* mutants corresponded to *clv3-10*.

**Transgenes**. The *R2D2* and *pDR5v2:3xVENUS-NLS* lines have been described in ref. [16]. *pDR5v2:tdTomato-Linker-NLS:trbcS* was transformed into heterozygous *wus-7* plants and L*er* control plants and activity patterns was scored in T1. A stable single insertion T3 line of *pDR5v2:ER-EYFP-HDEL:tAt4g24550* was used for transformation with *pCLV3:3xmCherry-NLS* and signals were scored in T1. For deGradFP the anti-GFP nanobody coding sequence (*NSlmb-vhhGFP4*)[24] was brought under control of the AlcR/AlcA system[46] and transformed into a stable *pWUS:WUS-linker-GFP wus* rescue line (GD44, described in ref. [5]), or an *pUBQ10: GFP-NLS* line as control. Experiments were performed in stable single insertion T3 lines. Similarly, the *pCLV3:AlcR/AlcA:CalS3m* line[5] was crossed to *pDR5v2:3xVENUS-NLS, pRPS5a:NLS-tdTomato* and F3 single insertion progeny was used for experiments. For generation of *MPΔ* we amplified a fragment of the *MP* cDNA, which codes for a protein that is truncated right before domain III (amino acids 1-794). For ectopic WUS induction lines *mCherry* was fused N-terminally to the ligand-binding domain of the rat glucocorticoid receptor (GR) and linked by (AAASAIAS[SG]11SAAA) to the *WUS* coding sequence under control of the *pUBQ10* promoter. A single insertion homozygous line was used for crossings, in RNA-seq, and ChIP-seq.

The *pHMG* promoter corresponds to 1347 bp upstream of the AT1g76110 locus. Most constructs were assembled using GreenGate cloning[47].

**Microscopy**. Confocal microscopy was carried out on an upright Nikon A1 Confocal with a CFI Apo LWD 25 × water immersion objective (Nikon Instruments) without coverslip as described in ref. [5]. 1 mg/ml DAPI was used for cell wall staining[5]. Pools of at least five plants were imaged for each time point.

**Image analysis**. Quantitative image analysis was done on isotropic image stacks using Fiji (v1.50b)[48], MorphoGraphX[49], ilastik[50], Matlab (Release 2014b, The MathWorks, Inc., United States) and KNIME[51]. All images for an experimental set were captured under identical microscope settings. Normalization was applied when settings had to be adjusted to correct for fluctuations over multiple days. MorphoGraphX analysis was performed according to standards defined in the user manual. Averaging and statistical analysis of signals across meristems was performed as follows: histograms of signal intensities along 100 central cross-sections per SAM (cross-sections rotated by 3.6 degrees successively) were measured by ImageJ standard function. Signals were centered for comparison between individuals. Signals +/− 12.5 μm around the SAM center were compared between treatment and control and tested for significance by Student's *t*-test. Distance from center with signal up to 120% of center background signal between treatment and control was determined and tested by Student's *t*-test.

To determine the center of an inflorescence meristem, 10 to 20 L1 cells located at the meristem summit were segmented using the carving workflow in ilastik. A sphere was fitted through the centroids of these cells using the least squared distances method. The sphere was superimposed on the original DAPI stained image volume to help identifying newly emerging flower primordia. Three points

marking the center of three young flower primordia were manually picked close to the sphere surface, projected onto the sphere and then used as seeds to perform a spheric voronoi tessellation (https://de.mathworks.com/matlabcentral/fileexchange/40989-voronoi-sphere). The point $P_{center}$ is equidistant to the three seed points and serves as a good approximation for the meristem center which is marked by the *pCLV3* stem cell reporter. The method was tested using image stacks of nine meristems containing cell walls stained by DAPI in one channel and the stem cell marker *pCLV3::mCherry-NLS* in the second channel. The computationally estimated meristem center and the one determined by *pCLV3: mCherry-NLS* expression in every case were in the range of one cell diameter. Further details and workflows are available on request.

For quantification of CLV3 and DR5 signals shown in Fig. 3 epidermal cell surfaces were segmented on 2.5D projections of DAPI-stained cell wall image volumes using MorphographX. DR5v2 and CLV3 signals from additional channels of the same stack were projected on to the same surface and respective mean signal intensity values along with the spatial coordinates for each cell were exported. After respective coordinate transformation distances to the central axis of the meristem (defined as described earlier) were computed for each cell. Only cells with dist <= 1/4*rad$_{sphere}$ were considered as stem cells and included in further analysis. Distances from the central axis, as well as mean signal intensities for each cell were min-max-normalized to compare DR5v5 and CLV3 mean signal intensities within the central stem cell domain of different meristems and different genotypes. Cells from four meristems were used for each genotype.

**In situ hybridization**. In-situ hybridizations were carried out as described in ref. [52]. Briefly, samples were fixed in PFA on ice, then transferred to a Leica Asp200 for automated tissue infiltration into Leica Histowax. Samples were sectioned at 8 μm and transferred to microscopy slides. After proteinase K digestion, samples were hybridized overnight at 55 °C with antisense RNA probes labeled with Digoxegenin (Roche) in a hybridization mix of 50% (deionized) formamide, 10% dextrane sulfate, 1× in situ Salts, 1× Denhardt's solution, 0.5 mg/mL tRNA. After washing with 2× SCC and 0.2× SSC at 55 °C, probes were detected by incubation with anti-Digoxegenin antibody linked to Alkaline Phosphatase (Roche) for 90 min. Visualization was carried out by NBT-BCIP after washing.

**ChIP-seq and RNA-seq**. All experiments were carried out on 5-day-old seedlings grown on 0.5 MS plates after 2 hours of either Dex or mock treatment in solution. Three gram of plant material was fixed in 100 mM sodium phosphate buffer, pH 7.0, 50 mM NaCl, 100 mM sucrose, and 0.33% formaldehyde for three times 10 min under vacuum. Nuclear extracts were prepared according to the INTACT protocol[53] and chromatin sheared in a sonication water bath to average fragment sizes of 200–400 bp. Cleared nuclear extracts were split in two and incubated each with 20 μl of RFP-Trap_A (ChromoTek, rta-100) for 4 h for immunoprecipitation. MinElute Reaction Cleanup Kit columns (Qiagen) were used for purification of the DNA fragments. Enrichment of specific DNA fragments was validated by qPCR at the ARR7 promoter region[26] by comparing immunoprecipitated DNA of WUS-GR and control samples. Libraries were generated for the *WUS-GR* and control ChIP using pooled DNA from 6 to 9 individual ChIP preparations. RNA-seq was carried out in biological triplicates. After careful benchmarking of our WUS-GR line, we find it to be the most potent and consistent tool for WUS induction to date, affording a much higher sensitivity for identifying transcriptional targets. In addition, the use of RFP-trap increased sensitivity of the ChIP assay. Consistently, we were able to identify 6740 genes whose chromatin region was bound by WUS. This compared to 136 regions we had previously identified by ChIP-chip[25], highlighting the increase in power. Previously identified direct targets, such as *ARR7, CLV1, KAN1, KAN2 AS2*, and *YAB3* (refs. [25–27]) were also picked up in our analysis. Because of the medium level ubiquitous expression of WUS, both RNA-seq and ChIP-seq capture the global regulatory potential of WUS. Since regulatory output of WUS is dependent on tissue context, targets identified here might not be relevant for all tissues. In addition, targets might be induced by WUS in one tissue and repressed in another, which cannot be resolved by this dataset. All genomic datasets are available under GEO accession: GSE122611.

**Bioinformatics**. Bioinformatic analysis was performed based on combination of R, command line and web-based bioinformatic tools. Alignments of all NGS data were done on a local Galaxy instance (v17.09)[54].

ChIP-seq data were mapped to TAIR10 genome by BWA aligner (v0.7.17)[55]. Identification of significant enrichment of ChIP-seq reads was performed using hiddenDomains (v3.0)[56] with the following window sizes: 100 bp for identification of WUS transcription factor binding sites and 500 bp for identification of histone modification marks. For further analysis, only peaks with posterior probability higher than 0.9 were used. All intervals which showed significant changes were transformed to GRanges[57] and reduced by merging overlapping or adjacent regions. Identified intervals showing changes in acetylation or methylation were removed from the dataset, if increase and decrease of the respective histone modification was found to be closer than 73 bp. Obtained intervals were transformed to BED files and annotated to TAIR10 genome using web-application PAVIS[58], using 2500 bp as upstream and 1000 bp as downstream regions.

Alignment of RNA-seq reads to TAIR10 genome by HISAT2 (v2.1.0)[59] and extraction of count matrices by featureCounts (v1.6.3)[60] were done on a local Galaxy instance. R bioconductor package DESeq2 (1.20.0)[61] was used for data analysis. One factor model was used for identification of genes transcriptionally regulated by WUS (design = ~Genotype), to obtain list of genes with interplay between genotype (WUS, WT) and treatment effect (TSA, Mock), a two-factor model with interaction term was used (design = ~Genotype + Treatment + Genotype:Treatment). Gene ontology analysis for genes responsive to WUS was carried out using topGO R package (v2.32.0) with all genes annotated to open chromatin[28] as background (Supplementary Table 1).

All results of NGS data analysis can be found in supplementary table.

**Reporting summary**. Further information on research design is available in the Nature Research Reporting Summary linked to this article.

## Data availability
Sequencing data is available under GEO accession GSE122611, processed data is contained in Supplementary Data 1 and 2. Quantification of all experiments is reported in the source data file. Biological resources described here are available from the corresponding author upon request.

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

## Acknowledgements

We thank Dolf Weijers for sharing R2D2 and DR5v2 resources before publication. This work was supported by the DFG through grants SFB1101 and SFB873 to J.U.L. and T.G. and by HFSP Grant RPG0054-2013 and ANR-12-BSV6-0005 grant to T.V. David Ibberson of the CellNetworks Deep Sequencing facility for Illumina sequencing. Computational analyses have been carried out on heiCLOUD provided by Heidelberg University Computing Centre.

## Author contributions

A.Me. performed in situ hybridizations, C.W. carried out imaging and analyses, J.F. established the WUS-GR line, G.U. and A. M. performed RNA-seq, O.E. performed bioinformatic analyses, K.B. and T.G. established the *pDR5v2:ER-EYFP-HDEL:tAt4g24550* line, C.G. made the *pCLV3:mCherry-NLS:tCLV3* construct, Z.Š., A.M. and Y.M. performed all other experiments. C.G.-A. and T.V. designed the TSA treatment of the SAM, Y.M., Z.Š., A.M. and J.U.L. designed all other experiments and wrote the paper with input from all other authors.

## Competing interests

The authors declare no competing interests.
