## [Peer Review File · Nature Communications]

Reviewers' comments:

Reviewer #1 (Remarks to the Author):

The authors have beautifully done a series of experiments to show that WUSCHEL acts as a rheostat on the auxin pathway to maintain apical stem cells. They first induced symplastic isolation through callose deposition at plasmodesmata of stem cells, which were shown earlier to induce their differentiation and showed that DR5v2 signal was induced in the central zone domain over time. Next, they showed that prolonged induction of a dominant negative auxin regulator BDL-D (IAA12) under CLV3 promoter caused meristem termination in WT and *clv3*. In contrast, expression of a potent positive signaling component, the auxin response factor ARF5/MONOPTEROS (MP), or its constitutively active form MP Δ did not cause changes in meristem size. These data clearly show that stem cells require auxin signaling for the maintenance, but are resistant to overactivation of the signaling pathway.

They further showed that after DEX treatment to pUBI10:mCherry-GR-linker-WUS transgene central auxin signaling minimum as well as the CLV3 domain expanded. By degrading WUS protein by adapting deGradFP technology and combined switchable stem cell specific expression of an anti-GFP nanobody with a pWUS:WUS-linker-GFP was rescue line, the opposite effects were observed, showing that WUS is required to rheostatically maintain auxin pathway to maintain stem cells.

Based on their sophisticated & sensitive ChIP protocol, they identified 6740 genomic regions bound by WUS. 2939 out of 6740 WUS bound chromatin regions showed acetylation changes, and 587 of the 1656 directly repressed WUS targets showed histone de-acetylation. Consistently, from the 1656 directly repressed genes, 938 were no longer responsive to WUS-GR induction when the deacetylase inhibitor TSA was present and roughly a third of them showed significant reduction in H3K9/K14 acetylation levels.

Finally, they grew plants segregating for *wus-7* on plates supplemented with auxin and observed twice as many terminated mutant seedlings on auxin plates compared to control plates, showing that WUS is the key factor to modulate auxin response in stem cells.

The experiments were basically carefully controlled, and the conclusion is solid. I have several concerns and comments mentioned below.

1) In Fig 1, please include pWUS:GFP reporter line.

2) In the experiment of induced symplastic isolation (Fig 2), it is not so clear what is the primarily causal for stem cell differentiation and in what timing symplastic isolation affects WUS mobility, auxin transport and signaling pathways. It would be good to observe reporter lines of R2D2 and PIN reporters as well as the WUS reporter in earlier time courses.

3) In Fig 2a-d, RPS5A ribosomal protein 5A was used as a control gene repressed during stem cell differentiation. The resolution of the figure is not so high. It will be good to show the higher magnification images and possible antagonistic expression of RPS5A and DR5v2. Further, RPS5A expression is not uniform even in the time course samples (Fig 2c shows much weaker RPS5A expression than others.)

4) A statistical description is absent in Fig 2g. Please performed ANOVA, followed by a post-hoc test.

5) The authors claimed "pCLV3:MP plants showed enhanced DR5v2 activity in stem cells (Fig.3c, d) demonstrating that ARF activity is indeed limiting for transcriptional output in wild-type." This is not so clear to me. Please quantify the data in 5-10 inflorescences.

6) In Fig 4j and k, it is good to show the PIN transporter and DII-VENUS reporters.

Reviewer #2 (Remarks to the Author):

The manuscript by Yanfei Ma et al. describes the results of extensive work studying the relationship between auxin and stem cells at the shoot apical meristem (SAM), and the mechanism by which WUS controls the auxin signaling to maintain the stem cell domain. The authors demonstrate that in the SAM an auxin output minimum correlates with apical stem cells and that this minimum is dependent on stem cell fate. The authors present data demonstrating that stem cells require active auxin signaling for their maintenance. Next they show that WUS monitors the auxin signaling in the stem cells and suggest that it acts via regulation of histone acetylation at genes from the auxin pathway.

In this study an impressive selection of methods and techniques have been used which yielded high quality images and figures some of which astonishing, yet some are not clearly presented (see below).

In general, this manuscript is overloaded with data from numerous experiments that attempt to present a comprehensive understanding of the WUS-auxin-stem cell circuit. However, regrettably many experiments are not well presented and some of the conclusion are not well supported. Some of their findings are novel but the experiment design is not detailed and it is difficult to evaluate. Also, it would be very helpful if the authors can discuss their findings in the context of previous reports.

Altogether I would suggest to break it into two papers with improved descriptions and discussions.

The following comments may help the authors improve their manuscript.

- In general, I see some conclusions that are not well supported. For example:

"Taken together, these results showed that WUS binds to and reduces transcription of the majority of genes involved in auxin signaling and response via de-acetylation of histones and thus is able to rheostatically maintain pathway activity in stem cells at a basal level"

1. I didn't find any data or list of genes bound by WUS and show reduction in expression that are involved in auxin signaling (you show in Fig 5 boxes that represent genes in the auxin pathway that are bound and response to WUS but nothing is said on the type of response).

2. I didn't see a list of genes from the auxin pathway that are bound by WUS and show reduction in histones acetylation and expression! (I also didn't see any list of genes related to auxin that are affected by TSA).

3. In general, I think that performing ChIP-seq on whole seedlings following induction of WUS function in all tissues (driven by strong promoter) requires that potential biases will be taken into account. Inducing WUS function in high level, might force the binding to G-Box sequence or to many of the WOX TF's targets, many of which might be genes of the auxin pathway (which are definitely not WUS targets).

Therefore, it is better to say potential targets of WUS and to discuss this bias.

- The readers will benefit from publishing the ChIP-seq data not only as a raw data (when the paper will be published) but as processed tables in the supplementary section. For example:

1. Supplementary Table 2: Response of genes with activities in auxin signalling to WUS. (there is a typo in signalling)

Beside the fact that this table lack a detailed description of the experimental design (Log fold change of what?, how many replicates etc. this data is missing also in the material and method section)

The authors do not provide the value of the RNA-seq for each gene (RPKM or other value). For me as a reader it is very difficult to evaluate your results since FC stand alone doesn't say much. It can be for example that the RPKM value for one treatment is very low and the other treatment is much lower such that the FC is very high but has no meaning!

2. " Consistently, from the 1656 directly repressed genes, 938 were no longer responsive to WUS-GR induction when TSA was present and roughly a third of them showed significant reduction in H3K9/K14 acetylation levels"

Please add tables with the relative value (Peak score, significance, RPKM etc) for all of the sets of genes (1656 repressed etc.).

- "MP mRNA expression had extended from the periphery into the central zone (Fig. 5b, c; Supplementary fig 5)"

NOT CONVINCING— 5b and 5c ---it seems to be a matter of exposure as the WUS-GFP section exhibit much strong signal at the periphery. S Fig 5: seems that the sections presented for the WUS degradation are not the middle one.

- Discussion.

With so many results you can extend you discussion and refer to some previous reports. I am giving only few examples but there are lots of related publish work that you can chose to add to your discussion:

A. You mention in the introduction the paper of Luo L et al (2018) but ignore their very relevant findings ("Our results provide a mechanistic framework for auxin control of shoot stem cell homeostasis and demonstrate how auxin differentially controls plant stem cell maintenance and differentiation")

B. In light of your ChIP-seq design (not on meristems), and as you don't provide a list of genes and you don't give the information about whether they are up or done regulated by WUS, you should discuss related paper like(Tian H et al Mol Plant. 2014 Feb;7(2):277-89)

"Here, we present unexpected evidence that WUSCHEL-RELATED HOMEBOX 5 (WOX5) transcription factor modulates expression of auxin biosynthetic genes in the quiescent center (QC) of the root and thus provides a robust mechanism for the maintenance of auxin response maximum in the root tip"

Specially as Laux T showed that WOX5 and WUS are interchangeable in stem cell control. (Sarkar AK Nature. 2007)

C. "Our findings provide evidence for the manner by which WUS specifies stem-cell identity: by affecting auxin responses (Negin et al 2017)

Some other remarks:

Many of the experiments, the legends and the tables are not detailed enough.

Few examples:

1. Fig 6 h: "n > 200 for each genotype and treatment"—that's mean that you analyzed 200 seedlings homozygous for wus-7? Or 200 seedling of segregated population, which mean that only 6 seedlings showed termination compared with Ler in which 4 seedlings exhibited termination? Please clarify!

In addition: In our lab we germinate Col seeds on 10 microM IAA and none of them grow normally. Please add pictures of the seedlings from this experiment to the supplement

2. "Neither of the 17 factors tested caused meristem phenotypes when expressed in stem cells (Fig. 2 and Table 1)"

First I couldn't find in Fig 2 such evidence (may be you meant Supp Fig 2)

Than Table 1 is totally not clear with no explanation to the table. This table to my understanding does not present the stated result!

3. Fig2 f) "Computational sphere fitting and identification of the central zone for fluorescence signal quantification"

It is not clear what this "Computational sphere" contributes. Since there is no detail it is unclear and useless.

4. "we locally blocked auxin output by our pCLV3:BDL-D transgene and observed stem cell termination phenotypes in almost all seedlings (n=30; Fig. 4c)"

The image in 4c does not show seedling – it is probably IFM (please correct).

Please state the number of plants (instead of almost)

5. Legend of Fig 5a is totally obscured, especially if you publish in a general journal – what data was used to generate the fig, please add that each box represents a gene in the indicated family (we don't have to guess it). The significance is not clear ("Within gene family tests are shown"?)

Last remark—there are many Typo and nomenclature that need to be fixed

For example:

- "Previously identified direct targets, such as ARR7, CLV1, KAN1, KAN2 AS2 and YAB3 (refs. 23-25)"

Change the refs 23-25 to the nature communication format

- "our profiling results were based on ectopic expression of WUS in non-stem cells"

WUS should be in *Italic*

- "This result suggested that also in fasciated SAMs of *clv3* mutants, ectopic WUS is sufficient to reduce auxin signaling"

WUS should be in *Italic*

(also: In the *clv3* mutant the WUS gene is upregulated not ectopically expressed)

Reviewer #3 (Remarks to the Author):

In the proposed work by Yanfei Ma et al. "WUSCHEL acts as a rheostat on the auxin pathway to maintain apical stem cells in Arabidopsis" the authors hypothesize the level of WUSCHEL within the apical pool of stem cells acts to modulate of the chromatin state of numerous auxin pathway genes, among others not addressed in the study. This repression, through the known recruitment of HISTONE DEACETYLASE COMPLEXES, maintains a state within the apical stem cell pool that allows low levels of auxin signaling while preventing differentiation, which balances with higher levels of auxin-mediated signaling on the flanks of the central zone to initiate new primordia. The topic on the mechanisms that balance maintenance of the stem cell pool with the acquisition of a differentiated cell state is highly relevant to both plant and animals. In addition, the role of auxin in mediating cellular differentiation and organ initiation in plants and its role in meristem function (primarily the root and vascular cambium) has been extensively studied. Despite this knowledge, relatively little is known about auxin function in the stem cells of the shoot outside its role in organ initiation. Therefore, the authors' study is timely and relevant.

There are a couple of clear advances presented in this work. First, the development of a method for the computational identification of the center of the SAM is an important advance. The authors provide good evidence based on the known expression pattern of the shoot apical stem cell marker

CLV3 that this computational method works well. Second, the authors provide another genome-wide dataset for WUSCHEL-regulated genes. While potentially very informative for the larger community of researchers in this field, the manner in which the data are presented is not that helpful. As referenced in the paper, this is the third such genome wide dataset this same group will have published regarding WUSCHEL regulation of target genes or its binding to target regions of the genome by chromatin immunoprecipitation. While an explanation is given as to why an additional dataset was required and some comparison to the previous published datasets is offered, only a very small subset of the data is presented in the text, i.e. specific to the auxin pathway. The paper would be greatly improved if a more comprehensive analysis of the dataset is shown (more than just the auxin pathway) highlighting what additional pathways or gene family are targets of WUSCHEL binding or regulated by WUSCHEL and what previously identified genes or pathways are confirmed in this dataset. As it stands right now in its current form, the genomics analysis feels highly selective and leaves the reader with many questions on what has been newly discovered in this dataset not previously known, especially in light of the figure given that a nearly 50-fold increase in number of WUSCHEL bound regions in the genome has been identified in this current study. In addition, no supporting validation for any of the referenced newly identified auxin related genes within the text is given in its present form. The only effort that seems to have been made in terms of validation of the new ChIP-seq dataset is a reference to quantification of a previously identified target WUSCHEL target, ARR7 from the materials and methods. As with any genome-wide ChIP-seq experiment, validation of binding is absolutely critical. As it stands right now, none of the proposed auxin pathway genes, shown binding by ChIP-seq, has supporting evidence by some other technique, such as quantification of enrichment when compared to control or identification of WUSCHEL binding element and subsequent binding analysis by electrophoretic mobility shift assay (EMSA).

The presentation of the new RNA-seq regulated genes, like the data for the ChIP-seq, is not clearly presented and highly selective. No validation by quantitative real-time PCR is offered for any of the genes presented in the manuscript. This is particularly problematic as the data are presented, as no information in the materials and methods is given as to what thresholds were used for the determination of a false discovery rate in the analysis. The authors need to provide this information in the text or supplement and validate that at least those genes of interest presented in the text are regulated by WUS in table 2. As a final note, in this reviewer's opinion the entire work would be better presented if the genomic analysis is shown first with the follow-up figures related to the work on auxin and stem cell function. It would also be beneficial if the information on the development of the computational methods for the identification of the SAM center is moved to the supplement with all supporting data and then referenced in the text.

The remaining main text figures two through four and supplemental figures two through five rely largely on the quantification of auxin response as read out by DR5v2 in the shoot apical meristem. The pattern of DR5 response in the shoot apical meristem has been characterized in numerous previous studies. One advance of this study is the analysis of acquisition of auxin response (as referenced by DR5v2) relative to expression of CLV3, which marks pluripotent stem cells in the SAM. The analysis done in figure 1F could be improved if not only the cells of the central zone are shown, but how the transition of high CLV3 expression to high auxin response (DR5v2) occurs spatially. From the images provided in figure D and E it is clear there is overlap in the expression domains for CLV3 and the DR5v2 response. If something could be said on the spatial determination of primordia initiation relative to CLV3 signal as the auxin response increases toward the periphery the paper, and the data of the subsequent figures could be strengthened, it would help to answer the open question of how much of the change in auxin response minimum is due to repression of the auxin signaling response or to expansion of the CLV3 marked stem cells in that central domain shown in figure 1F, in the subsequent data presented in figures two through four.

In figures two through four and supplemental figures 2 through 5 I will outline my comments below as major and minor points.

Major Point 1: The analysis of the data presented herein does not take into account several relevant previously published works. In previously published work either overexpression of CLV3 throughout the SAM or a WUS RNAi throughout the SAM caused meristem termination (1). Either

gain of CLV3 function or loss of WUSCHEL function results in no activation of DR5 in the center of the SAM, while termination occurs. In this current study, the authors show that only reducing WUSCHEL protein in the L1/L2, by an anti-GFP nanobody, is sufficient to alter the DR5v2 distribution in the SAM, shifting it toward the center. In addition, this current work ignores another figure in the Yadav et al. Development 2010 publication where activation of WUS in the CLV3 expressing cells results in a temporal and spatial expansion in CLV3 expression over time. Importantly, those cells at the periphery of the central zone that had acquired primordium fate, prior to WUS activation, did not express CLV3 during the 120-hour time course. This implies that once primordial fate is acquired those cells are locked into that fate. However, the zone of CLV3 expands radially, around those CLV3 negative cells, while more primordia are formed on the flanks of the enlarged meristem compared to mock treated shoot apical meristems. In light of these results, an explanation for the results shown in Figure 4 panels D and E, where WUS-GR is activated and the expression minimum for DR5v2 is expanded, could simply be due to a cell fate respecification shift toward the CLV3 expression stem cell fate, prior to acquisition of primordial cell fate at the border where cells transiently express both CLV3 and display DR5v2 signal. In fact, evidence for this can be supported by the work of Reddy and Meyerowitz (2), where reduction of CLV3 peptide in the central zone resulted in a respecification of CLV3 non-expressing cells at the border of the central zone back to CLV3 expressing stem cells within a 24 hour window. The authors appear to attempt to address this in supplementary figure 3 and show the CLV3 domain is increased after WUS-GR activation but this is nothing like what is shown for the expansion of CLV3 in other references listed above. If this slight increase is what is observed this would contradict previous studies. For the authors to demonstrate this they would have to show quantification of the CLV3 domain size in great numbers of mock and dexamethasone treated WUS-GR shoot apices. Furthermore, given that the methodologies are different in these referenced studies, the overall experimental approaches should arrive at a similar end (at certain time points), how the authors of this current work reconcile their current results with these published works needs to be addressed.

In Bhatia et al. (3), the authors claim that long term induction or expression of a constitutive active form of MPΔ, which lacks domain III and IV, results in transgene silencing. The authors here need to ensure that in the results of Figure 3 and supplementary figure 3 the transgene is active. Also, in Bhatia et al. it is shown that activation of this MP truncated form causes a halt of organ initiation and the authors propose that MP is not permissive but instructive for organ initiation by convergence of PIN1 polarity. If this were the case, I would expect the plants of the pCLV3::MPΔ genotype to show a similar phenotype to that shown in Bhatia et al. where organ initiation is blocked and cells at the apex proliferate into an undifferentiated mass in seedlings where MPΔ is activated. Even though there is a difference in promoters used in these studies, UBQ10 vs CLV3, given that CLV3 is expressed during early embryo development when the apical stem cells are specified, I would assume that expression of MPΔ would give a similar overproliferation phenotype where differentiation on the flanks was terminated. Given the issues of transgene silencing reported by Bhatia et al. the authors need to show evidence that their transgene is expressed at the appropriate developmental time analyzed in figure 3/supplemental figure 2 in the shoot apical meristem. The fact that expression of MPΔ from the At1g76110 HMG promoter results in production of numerous flowers while expression from the CLV3 promoter has little affect the authors need to show that 1.) indeed the stem cell pool is not distributed in the pCLV3:MPD experiment and 2.) The stem cells are maintained in the HMG:MPD experiment while peripheral zone cells are impacted by the presence of MP activity. In addition, the reference given for the MPD is not correct. The reference given characterizes the genomic loci for MP but does not state anything about the removal of domains III and IV for the generation of MPD. The authors should use Ckurshumova W., Krogan N.T., Marcos D., Caragea A.E., Berleth T. Irrepressible, truncated auxin response factors: natural roles and applications in dissecting auxin gene regulation pathways. Plant Signal. Behav. 2012; 7: 1027-1030.

This work also does not reference or address the work of Wu et al. (4), which demonstrates MP works through chromatin remodeling to induce floral primordium fate. Can the authors here provide an explanation of why when a constitutively active form of MP is expressed in the CZ there is not a switch to the PZ cell fate? In this current work WUSCHEL is proposed to work through the

HDAC-mediated repression of auxin pathway genes, while the work of Wu et al. would say the SWI/SNF complex would turn on primordium-specific genes. Is this due to WUSCHEL having a dominant role in the stem cells to maintain a repressive chromatin state? A simple test could be done to address if there is a shift in acetylation state of the chromatin in both the pCLV3::MPΔ SAM or the pHMG::MPΔ SAM. Given that primordium production is increased in only the pHMG::MPΔ line you might expect the chromatin isolated from the SAMs of these plants to have a higher percentage of acetylation.

Major point 2: In the use of the pCLV3:iCalSm the mechanism of loss of stem cell identity is not clear. Previous work in reference 5 gave evidence that lack of WUSCHEL movement into these cells could be the mechanism. However, in this figure only the activation of DR5v2 is shown. To make this claim definitively, the authors need to show how the expression of CLV3 changes upon activation of the iCalSm. In reference 5, the time frame presented for meristem termination is 72 hours. This study shows a time course of 120 hours where the meristem appears to still be functioning despite the appearance of DR5v2 in the center of the SAM. It is also not clear why RPS5a is used as the marker for stem cells, when CLV3 expression would be much more appropriate. Yes, RPS5a is expressed in meristematic tissue, but it is also expressed in many other cell types. The authors give no explanation on why this particular promoter is used in lieu of a more traditional stem cell marker. If the closure of plasmodesma in the stem cells does result in their differentiation one point that is not addressed by these experiments is how does the auxin level change given the sharp increase in DR5v2 activity shown. One would expect that the levels of auxin would increase and a change in the R2D2 ratio would be observed. If these data were provided it would further strengthen the authors conclusions give the issues listed above. As stated early, given what is presented in figure 1, it would seem that for all the subsequent experiments the analysis of DR5v2 signal, if done in tandem with CLV3 expression, the concerns outlined above would be answered. Given that in figure 3/supplemental figure 2 the authors ectopically express MP in the CZ or inhibit auxin signaling by the use of a dominant active BDL-D and there no stem cell loss is observed. In addition, when CLV3 is overexpressed in the SAM, where stem cells terminate, KAN1 expression is seen more toward the center of the SAM, indicating the loss of stem cells in the center is coupled to peripheral zone markers moving toward the central region of the SAM (5). This would contradict what is shown in Figure 2 of this current work, where symplastically isolating the CLV3 expressing cells, resulting in their differentiation, does not show any observed disruption in the zonation of the SAM, just appearance of DR5v2 in the center. However, no other specific marker of the SAM stem cell niche is analyzed in these experiments.

Major point 3: From the results shown and the discussion at the end of the text, the authors would like to claim that the expression of dominant negative auxin regulators result in SAM arrest, and have used a dominant active form of BODENLOS (BDL-D) to arrive at this claim. However, BDL is not expressed in the center of the SAM but at the periphery according to previously published in situ hybridization work. This would support their hypothesis. However, in the embryo it is associated with specification of lower basal tier of the embryo, not the apical tier (6). In figure 3, supplementary figure 2, and figure 4 overexpression of the dominant negative form of BDL (BDL-D) from the CLV3 promoter is shown to enlarge the auxin minimum transiently and cause meristem termination if expressed constitutively. However, there is no determination of how BDL-D impacts CZ identity through the analysis of CLV3 expression. The concern is that the expression of BDL-D not just impacts the auxin signaling status but possibly the cell fate when expressed throughout the CLV3 cells, this would be especially relevant in figure 3 and 4 when expressed constitutively from the CLV3 promoter that is activated fairly early in the heart stage. If the authors demonstrated that the CLV3 expression is still present after 24 hours or at some point during the constitutive expression experiments followed by stem cell depletion over time this would strongly support their hypothesis. Furthermore, these results imply that a subset of AUX/IAA genes are repressed in the central zone for the maintenance of the stem cell pool. Given the authors generate data that identifies several AUX/IAA targets as regulated or bound by WUS, which fit the criteria for the mechanism the authors propose, their hypothesis would be greatly strengthened if they could show some of these repressive AUX/IAA genes are turned on, or are

upregulated, in the absence of WUS activity, or are repressed when WUSCHEL is expressed in the PZ cells. The only data presented is regarding the expression of either MP and TIR1, where it is not all that clear there is a significant change. To support this is central to their claim that WUS acts to balance auxin signaling in the stem cells of the SAM.

Major point 4: In reference 5 of the main text, these same authors stated only a C-terminal tagged WUS protein with a 30 amino acid linker can complement the *wus* null phenotype. Did the authors characterize this N-terminal tagged version for complementation of the *wus* phenotype? There is no information regarding this point in the text and reading the materials and methods would suggest that this transgene was transformed into wild type Columbia. This is a key point because in their previous report they indicated N-terminal fusion with WUSCHEL lack of complementation implying lack of WUSCHEL function. This line is used for the ChIP-seq, RNA-seq, and imaging experiments of figure 4 and supplementary figure 3.

Minor point 1: There are no values associated with the y-axis in figure 2G.

Minor point 2: In the explanation of the data shown in figure 2 how the callose synthase is expressed is given as reference 5 in the main text. In the materials and methods, it is indicated that the CLV3 promoter was used to drive the expression of callose synthase (*iCalSm*). I would be informative to those not familiar with this field if an explicit statement is given the main text or figure legend that the callose synthase was expressed from the stem cell specific CLV3 promoter.

Minor point 3: The activation of DR5v2 in the center of the SAM would be much clearer if the data were presented as a Z/Y or Z/X slice in addition to the projection shown. Is this DR5v2 activation in the L1-L3/4, the primary domain of CLV3 expression, or broader in the deeper layers given that the authors have previously shown that cells enlarge after 72 hours in the L3/L4 of the SAM signifying differentiation.

Minor point 4: In all figures showing time course data it is not clear in either the main text or in the figure legend if the same shoot apical meristem is being imaged and what was the frequency of imaging.

Minor point 5: The alleles for *clv3* used in this study are not provided and need to be.

Minor point 6: What is meant by nearly all in figure 4C? Please provide numbers to support this claim. Also, the text here indicates seedlings but the image in figure 4 clearly shows a plant with a stem and what appears to be a cauline leaf? This does not look like a seedling. If this is a seedling do they flower early and do the SAMs terminate, after producing few flowers at the seedling stage but still produce a stem that elongates? This would suggest some pool of apical stem cells present to produce the stem tissue?

Minor point 7: In figure 4 panels G and H, can the authors address why the WUS-GFP signal is decreased in the basal aspects of the WUS expression domain (toward the pith at the bottom of the rib meristem) in addition to the cell of L1/L2? The expression domain of the CLV3 gene extends at least to the L3 and realistically to the L4. Why is there reduction in GFP signal below the cells of the L3/L4 while maintained in these cells? Is this due to a feedback mechanism that increases expression from the WUS promoter?

Minor point 8: In the graph shown in Figure 2G there are no values shown on the y-axis. Please provide values and statistical analysis. In the graph shown in figure 4F no X-axis is shown. For clarity it would be good to have an X-axis with a low value starting at 0.

Minor point 9: The data shown in main text figure 4L for the *wus-7* results would be more appropriate if they were moved to supplementary figure 4 where the images for DR5v2 distribution are shown in the *wus-7* background.

For figure 5 panels B and C, the 24 mock control needs to be shown. If one is to compare the mock treatment in the UBQ10::WUS-GR experiment this really is no change in the distribution of the signal.

Minor point 10: In the main text and materials and methods the WUS-GR transgene is listed as pUBI10, I believe this should be pUBQ10.

Minor point 11: The significant digits in Table 2 need to be trimmed.

Minor point 12: In the materials and methods the concentration for induction of WUS-GR is given but it is not listed for the BDL-D-GR experiments in Figure 3.

Minor point 13: Please provide the details for the generation of the MP Δ . Does the construct in this work correspond to the same construct reference in major point 1 (3).

1. Yadav, R. K., Tavakkoli, M. & Reddy, G. V. WUSCHEL mediates stem cell homeostasis by regulating stem cell number and patterns of cell division and differentiation of stem cell progenitors. *Development* 137, 3581 (2010).
2. Reddy, G. V. & Meyerowitz, E. M. Stem-Cell Homeostasis and Growth Dynamics Can Be Uncoupled in the Arabidopsis Shoot Apex. *Science* 310, 663 (2005).
3. Bhatia, N. et al. Auxin Acts through MONOPTEROS to Regulate Plant Cell Polarity and Pattern Phyllotaxis. *Curr. Biol.* 26, 3202–3208 (2016).
4. Wu, M.-F. et al. Auxin-regulated chromatin switch directs acquisition of flower primordium founder fate. *eLife* 4, e09269 (2015).
5. Yadav, R. K. et al. Plant stem cell maintenance involves direct transcriptional repression of differentiation program. *Mol. Syst. Biol.* 9, 654 (2013).
6. Hamann, T., Mayer, U. & Jurgens, G. The auxin-insensitive bodenlos mutation affects primary root formation and apical-basal patterning in the Arabidopsis embryo. *Development* 126, 1387 (1999).

We would like to thank all reviewers for their careful and constructive evaluation of our manuscript. Following their advice, we have carried out new experiments and quantifications and have now prepared a heavily revised version which includes 11 new panels in the main figures, 4 new supplementary figures, one new and two modified supplementary tables, a huge CSV file with 18 new spreadsheets giving direct access to processed data for all genomic datasets, an interactive HTML file that allows intuitive mining of our genomic data, as well as a substantially revised text with additional references.

Reviewer #1 :

The authors have beautifully done a series of experiments to show that WUSCHEL acts as a rheostat on the auxin pathway to maintain apical stem cells. They first induced symplastic isolation through callose deposition at plasmodesmata of stem cells, which were shown earlier to induce their differentiation and showed that DR5v2 signal was induced in the central zone domain over time. Next, they showed that prolonged induction of a dominant negative auxin regulator BDL-D (IAA12) under CLV3 promoter caused meristem termination in WT and *clv3*. In contrast, expression of a potent positive signaling component, the auxin response factor ARF5/MONOPTEROS (MP), or its constitutively active form MP Δ did not cause changes in meristem size. These data clearly show that stem cells require auxin signaling for the maintenance, but are resistant to overactivation of the signaling pathway.

They further showed that after DEX treatment to pUBI10:mCherry-GR-linker-WUS transgene central auxin signaling minimum as well as the CLV3 domain expanded. By degrading WUS protein by adapting deGradFP technology and combined switchable stem cell specific expression of an anti-GFP nanobody with a pWUS:WUS-linker-GFP was rescue line, the opposite effects were observed, showing that WUS is required to rheostatically maintain auxin pathway to maintain stem cells.

Based on their sophisticated & sensitive ChIP protocol, they identified 6740 genomic regions bound by WUS. 2939 out of 6740 WUS bound chromatin regions showed acetylation changes, and 587 of the 1656 directly repressed WUS targets showed histone de-acetylation. Consistently, from the 1656 directly repressed genes, 938 were no longer responsive to WUS-GR induction when the deacetylase inhibitor TSA was present and roughly a third of them showed significant reduction in H3K9/K14 acetylation levels.

Finally, they grew plants segregating for *wus-7* on plates supplemented with auxin and observed twice as many terminated mutant seedlings on auxin plates compared to control plates, showing that WUS is the key factor to modulate auxin response in stem cells.

The experiments were basically carefully controlled, and the conclusion is solid. I have several concerns and comments mentioned below.

RESPONSE:

Thank you very much for your kind assessment and the useful comments to improve our manuscript.

1) In Fig 1, please include pWUS:GFP reporter line.

RESPONSE:

Since in wild type meristems auxin signaling occurs mostly within the L1 layer, we focused our analyses on the CLV3 promoter. When imaging from the top, WUS and CLV3 reporters perfectly overlap (see for example Gailloch et al. eLife DOI: [10.7554/eLife.30135](https://doi.org/10.7554/eLife.30135) Figure

2—figure supplement 3). Because of this and the spatial separation of WUS and auxin reporters, adding pWUS-GFP would not add any information.

2) In the experiment of induced symplastic isolation (Fig 2), it is not so clear what is the primarily causal for stem cell differentiation and in what timing symplastic isolation affects WUS mobility, auxin transport and signaling pathways. It would be good to observe reporter lines of R2D2 and PIN reporters as well as the WUS reporter in earlier time courses.

RESPONSE:

Thanks for bringing up these important points. The effect of closing plasmodesmata on WUS mobility and thus on stem cells has been carefully analyzed and published in Daum et al. (PNAS 2014). We now mention in the text that stem cell differentiation is caused by restriction of WUS protein mobility and refer to the timing reported in Daum et al. We agree that it would be interesting to observe R2D2 and PIN1 in these plants, but since for this manuscript the symplastic isolation is just a tool to trigger stem cell differentiation, we feel this is very much out of scope for the present manuscript.

3) In Fig 2a-d, RPS5A ribosomal protein 5A was used as a control gene repressed during stem cell differentiation. The resolution of the figure is not so high. It will be good to show the higher magnification images and possible antagonistic expression of RPS5A and DR5v2. Further, RPS5A expression is not uniform even in the time course samples (Fig 2c shows much weaker RPS5A expression than others.)

RESPONSE:

We apologize for the poor resolution in our manuscript PDF. The apparently weaker RPS5a signal was caused by a much stronger cell wall stain in the old panel c. To overcome this, we have now selected a different image for the 72 hours timepoint, which is much more similar to the other timepoints. In addition, we have carefully quantified both RPS5a and DR5v2 signals and used RPS5a for normalization.

4) A statistical description is absent in Fig 2g. Please performed ANOVA, followed by a post-hoc test.

RESPONSE:

Thank you for bringing up this important point. We have now analyzed the data by ANOVA and post-hoc test and prepared a new figure panel showing raw data, box plots, as well as post-hoc p values.

5) The authors claimed “pCLV3:MP plants showed enhanced DR5v2 activity in stem cells (Fig.3c, d) demonstrating that ARF activity is indeed limiting for transcriptional output in wild-type.” This is not so clear to me. Please quantify the data in 5-10 inflorescences.

RESPONSE:

We have now repeated the experiment and independently confirmed the results shown in Fig. 3. In addition, we have carefully quantified the data using kernel regression to visualize the dependence of fluorescence signal on distance from SAM centre for both CLV3 and DR5 reporters in wild type, CLV3-MP and CLV3-MP Δ . The results, including 95% confidence intervals, are now shown in Fig. 3 m and n.

6) In Fig 4j and k, it is good to show the PIN transporter and DII-VENUS reporters.

RESPONSE:

We agree with the reviewer that it would be good to know about PIN and R2D2 in the WUS depletion background. However, for the purpose of this manuscript auxin signaling output is by far the most important read out in this experiment. Importantly, all GFP variants in the cytoplasm or nucleus (such as PIN or R2D2) would be targets of the anti-GFP nanobody and therefore degraded. Hence, the suggested experiment is unfortunately not feasible.

Reviewer #2 :

The manuscript by Yanfei Ma et al. describes the results of extensive work studying the relationship between auxin and stem cells at the shoot apical meristem (SAM), and the mechanism by which WUS controls the auxin signaling to maintain the stem cell domain. The authors demonstrate that in the SAM an auxin output minimum correlates with apical stem cells and that this minimum is dependent on stem cell fate. The authors present data demonstrating that stem cells require active auxin signaling for their maintenance. Next they show that WUS monitors the auxin signaling in the stem cells and suggest that it acts via regulation of histone acetylation at genes from the auxin pathway.

In this study an impressive selection of methods and techniques have been used which yielded high quality images and figures some of which astonishing, yet some are not clearly presented (see below).

In general, this manuscript is overloaded with data from numerous experiments that attempt to present a comprehensive understanding of the WUS-auxin-stem cell circuit. However, regrettably many experiments are not well presented and some of the conclusion are not well supported. Some of their findings are novel but the experiment design is not detailed and it is difficult to evaluate. Also, it would be very helpful if the authors can discuss their findings in the context of previous reports.

Altogether I would suggest to break it into two papers with improved descriptions and discussions.

RESPONSE:

Thank you very much for your encouragement and your valuable comments. We have performed a number of additional bioinformatic analyses, made available all genomic data that were requested and created an interactive HTML tool to visualize and mine our rich datasets. Since for the purpose of this manuscript the genomic data serve as a molecular phenotype and to address a specific question based on the in vivo experiments, we prefer to keep the data in a single manuscript. We have improved the description and presentation of the genomic data as you requested.

The following comments may help the authors improve their manuscript.

- In general, I see some conclusions that are not well supported. For example:
"Taken together, these results showed that WUS binds to and reduces transcription of the majority of genes involved in auxin signaling and response via de-acetylation of histones and thus is able to rheostatically maintain pathway activity in stem cells at a basal level"

1. I didn't find any data or list of genes bound by WUS and show reduction in expression that are involved in auxin signaling (you show in Fig 5 boxes that represent genes in the auxin pathway that are bound and response to WUS but nothing is said on the type of response).

RESPONSE:

Thank you for bringing up this important point. This information can be found in Supplementary Table 2

2. I didn't see a list of genes from the auxin pathway that are bound by WUS and show reduction in histones acetylation and expression! (I also didn't see any list of genes related to auxin that are affected by TSA).

RESPONSE:

We have now included this information in the new Supplementary Table 3. In addition, we developed an interactive HTML file that allows all experimental intersections to be visualized and all genes to be listed and mined. We very much hope that this effort allows all readers to access the data in an easy and intuitive fashion.

3. In general, I think that performing ChIP-seq on whole seedlings following induction of WUS function in all tissues (driven by strong promoter) requires that potential biases will be taken into account. Inducing WUS function in high level, might force the binding to G-Box sequence or to many of the WOX TF's targets, many of which might be genes of the auxin pathway (which are definitely not WUS targets).

Therefore, it is better to say potential targets of WUS and to discuss this bias.

RESPONSE:

Thank you for bringing up this important point. As explained above, we are fully aware of the limitations of our experimental approach and have given substantial explanation in the methods section. In addition, we have now revised the relevant section in the text to highlight the fact that WUS is ectopically expressed at moderate levels and that we record the entire repertoire of potential WUS binding events.

• The readers will benefit from publishing the ChIP-seq data not only as a raw data (when the paper will be published) but as processed tables in the supplementary section. For example:

1. Supplementary Table 2: Response of genes with activities in auxin signalling to WUS.
(there is a typo in signalling)

RESPONSE:

We have now included all processed data in an excel sheet to give readers direct access to all our genomic experiments. In addition, all relevant can be interrogated using our interactive HTML tool.

Beside the fact that this table lack a detailed description of the experimental design (Log fold change of what?, how many replicates etc. this data is missing also in the material and method section)

The authors do not provide the value of the RNA-seq for each gene (RPKM or other value). For me as a reader it is very difficult to evaluate your results since FC stand alone doesn't say much. It can be for example that the RPKM value for one treatment is very low and the other treatment is much lower such that the FC is very high but has no meaning!

RESPONSE:

We fully agree with this reviewer some essential information was lacking and have now included the relevant details in the methods and processed datasets in Supplementary file 1. Information on replication and pooling were already described out in the original version of the methods.

2. " Consistently, from the 1656 directly repressed genes, 938 were no longer responsive to WUS-GR induction when TSA was present and roughly a third of them showed significant reduction in H3K9/K14 acetylation levels"

Please add tables with the relative value (Peak score, significance, RPKM etc) for all of the sets of genes (1656 repressed etc.).

RESPONSE:

We have now included all relevant data in Supplementary File 1 to give readers direct access to all our genomic experiments. In addition, all relevant can be interrogated using our interactive HTML tool.

• "MP mRNA expression had extended from the periphery into the central zone (Fig. 5b, c; Supplementary fig 5)"

NOT CONVINCING— 5b and 5c ---it seems to be a matter of exposure as the WUS-GFP section exhibit much strong signal at the periphery. S Fig 5: seems that the sections presented for the WUS degradation are not the middle one.

RESPONSE:

Our intention is to show the true variation in this experiment that stems from differences in nanobody activity and in situ hybridization. All images were carefully selected from serial sections and to the best of our knowledge represent central parts of the SAM.

• Discussion.

With so many results you can extend you discussion and refer to some previous reports. I am giving only few examples but there are lots of related publish work that you can chose to add to your discussion:

A. You mention in the introduction the paper of Luo L et al (2018) but ignore their very relevant findings ("Our results provide a mechanistic framework for auxin control of shoot stem cell homeostasis and demonstrate how auxin differentially controls plant stem cell maintenance and differentiation")

B. In light of your CHIP-seq design (not on meristems), and as you don't provide a list of genes and you don't give the information about whether they are up or done regulated by WUS, you should discuss related paper like(Tian H et al Mol Plant. 2014 Feb;7(2):277-89) "Here, we present unexpected evidence that WUSCHEL-RELATED HOMEBOX 5 (WOX5) transcription factor modulates expression of auxin biosynthetic genes in the quiescent center (QC) of the root and thus provides a robust mechanism for the maintenance of auxin response maximum in the root tip"

Specially as Laux T showed that WOX5 and WUS are interchangeable in stem cell control. (Sarkar AK Nature. 2007)

C. "Our findings provide evidence for the manner by which WUS specifies stem-cell identity: by affecting auxin responses (Negin et al 2017)

RESPONSE: We have included the suggested references in the discussion.

Some other remarks:

Many of the experiments, the legends and the tables are not detailed enough.

Few examples:

1. Fig 6 h: "n > 200 for each genotype and treatment"—that's mean that you analyzed 200 seedlings homozygous for *wus-7*? Or 200 seedling of segregated population, which mean that only 6 seedlings showed termination compared with Ler in which 4 seedlings exhibited termination? Please clarify!

In addition: In our lab we germinate Col seeds on 10 microM IAA and none of them grow normally. Please add pictures of the seedlings from this experiment to the supplement

RESPONSE: Thank you for pointing out this unclear description: The percentages refer to the segregating population, which was tested in two independent experiments: on mock 42 of 720 seedlings segregating *wus-7* terminated; on IAA 82 of 617 seedlings segregating *wus-7* terminated. We genotyped all arrested plants – all of them were homozygous for *wus-7*. We have now modified Fig. 6 h to show the results from both experiments with the exact n and used give chi-square test derived p-values to compare the behaviour of Ler and *wus-7* populations. We have also included all raw numbers in the figure legend.

To address the concerns about responses of wild type Ler seedlings on auxin plates, we have now performed additional experiments at 1 μ M, 5 μ M and 20 μ M IAA. While we do observe enhanced termination of *wus-7* on 10 and 20 μ M, wild type Ler seedlings do not exhibit SAM arrest even at 20 μ M. We have now prepared a new supplementary figure 9 to show the phenotypic classes of these seedlings, as well as to present these new datasets.

2. "Neither of the 17 factors tested caused meristem phenotypes when expressed in stem cells (Fig. 2 and Table 1)"

First I couldn't find in Fig 2 such evidence (may be you meant Supp Fig 2)

Than Table 1 is totally not clear with no explanation to the table. This table to my understanding does not present the stated result!

RESPONSE:

We are very sorry for the confusion. We should have referred to Fig. 3 and Table 1. We have corrected the mistake and added more information to Table 1.

3. Fig2 f) "Computational sphere fitting and identification of the central zone for fluorescence signal quantification"

It is not clear what this "Computational sphere" contributes. Since there is no detail it is unclear and useless.

RESPONSE:

The sphere fitting allows us to identify the central zone without genetic marker and thus is of great importance for some of our experiments. The entire image analysis pipeline is described in the methods section in some detail.

4. "we locally blocked auxin output by our pCLV3:BDL-D transgene and observed stem cell termination phenotypes in almost all seedlings (n=30; Fig. 4c)"

The image in 4c does not show seedling – it is probably IFM (please correct).

Please state the number of plants (instead of almost)

RESPONSE:

We have repeated the experiment and have observed termination in all 26 T1 plants. In our first experiment, we had analysed 30 T1 plants, all of which terminated, almost all during seedling stage. We have chosen the particular plant for figure 4c that terminated only at inflorescence stage, since it shows the fasciated stem typical for *clv3* mutants. We have now changed the text to reflect this and the new numbers.

5. Legend of Fig 5a is totally obscured, especially if you publish in a general journal — what data was used to generate the fig, please add that each box represents a gene in the indicated family (we don't have to guess it). The significance is not clear ("Within gene family tests are shown"?)

RESPONSE: Thank you for pointing this out. We have modified Fig 5 a to clarify how enrichment was tested and have substantially expanded the legend to address this issue.

Last remark—there are many Typo and nomenclature that need to be fixed

For example:

- "Previously identified direct targets, such as ARR7, CLV1, KAN1, KAN2 AS2 and YAB3 (refs. 23-25)"

Change the refs 23-25 to the nature communication format

- "our profiling results were based on ectopic expression of *WUS* in non-stem cells"

WUS should be in Italic

- "This result suggested that also in fasciated SAMs of *clv3* mutants, ectopic *WUS* is sufficient to reduce auxin signaling"

WUS should be in Italic

(also: In the *clv3* mutant the *WUS* gene is upregulated not ectopically expressed)

RESPONSE:

We thank the reviewer for carefully reading our manuscript and have carefully checked for the issues pointed out. We disagree with regards to *WUS* expression in *clv3* mutants, since *WUS* RNA is clearly expressed in a much wider domain, as well as in L2, where it is usually never found.

Reviewer #3:

In the proposed work by Yanfei Ma et al. "WUSCHEL acts as a rheostat on the auxin pathway to maintain apical stem cells in Arabidopsis" the authors hypothesize the level of WUSCHEL within the apical pool of stem cells acts to modulate of the chromatin state of numerous auxin pathway genes, among others not addressed in the study. This repression, through the known recruitment of HISTONE DEACETYLASE COMPLEXES, maintains a state within the apical stem cell pool that allows low levels of auxin signaling while preventing differentiation, which balances with higher levels of auxin-mediated signaling on the flanks of the central zone to initiate new primordia. The topic on the mechanisms that balance maintenance of the stem cell pool with the acquisition of a differentiated cell state is highly relevant to both plant and animals. In addition, the role of auxin in mediating cellular differentiation and organ initiation in plants and its role in meristem function (primarily the root and vascular cambium) has been extensively studied. Despite this knowledge, relatively little is known about auxin function in the stem cells of the shoot outside its role in organ initiation. Therefore, the authors' study is timely and relevant.

There are a couple of clear advances presented in this work. First, the development of a method for the computational identification of the center of the SAM is an important advance. The authors provide good evidence based on the known expression pattern of the shoot apical stem cell marker CLV3 that this computational method works well. Second, the authors provide another genome-wide dataset for WUSCHEL-regulated genes. While potentially very informative for the larger community of researchers in this field, the manner in which the data are presented is not that helpful. As referenced in the paper, this is the third such genome wide dataset this same group will have published regarding WUSCHEL regulation of target genes or its binding to target regions of the genome by chromatin immunoprecipitation. While an explanation is given as to why an additional dataset was required and some comparison to the previous published datasets is offered, only a very small subset of the data is presented in the text, i.e. specific to the auxin pathway.

RESPONSE:

We thank this reviewer for the careful evaluation of our work and the insightful comments.

The paper would be greatly improved if a more comprehensive analysis of the dataset is shown (more than just the auxin pathway) highlighting what additional pathways or gene family are targets of WUSCHEL binding or regulated by WUSCHEL and what previously identified genes or pathways are confirmed in this dataset. As it stands right now in its current form, the genomics analysis feels highly selective and leaves the reader with many questions on what has been newly discovered in this dataset not previously known, especially in light of the figure given that a nearly 50-fold increase in number of WUSCHEL bound regions in the genome has been identified in this current study. In addition, no supporting validation for any of the referenced newly identified auxin related genes within the text is given in its present form.

RESPONSE:

Our manuscript is not primarily about another genome wide dataset of WUS, instead we try to elucidate why stem cells have a differential response to auxin. Hence, our analysis is highly selective by choice not to confuse the reader. We agree that the data is of high value for many readers and have therefore performed a number of additional bioinformatic analyses, assembled a large excel sheet with all processed data and many useful gene-lists and importantly have created an interactive HTML tool that allows all readers to visualize and mine our data.

The only effort that seems to have been made in terms of validation of the new ChIP-seq dataset is a reference to quantification of a previously identified target WUSCHEL target, ARR7 from the materials and methods. As with any genome-wide ChIP-seq experiment, validation of binding is absolutely critical. As it stands right now, none of the proposed auxin pathway genes, shown binding by ChIP-seq, has supporting evidence by some other technique, such as quantification of enrichment when compared to control or identification of WUSCHEL binding element and subsequent binding analysis by electrophoretic mobility shift assay (EMSA).

The presentation of the new RNA-seq regulated genes, like the data for the ChIP-seq, is not clearly presented and highly selective. No validation by quantitative real-time PCR is offered for any of the genes presented in the manuscript. This is particularly problematic as the data are presented, as no information in the materials and methods is given as to what thresholds were used for the determination of a false discovery rate in the analysis. The authors need to provide this information in the text or supplement and validate that at least those genes of interest presented in the text are regulated by WUS in table 2. As a final note, in this reviewer's opinion the entire work would be better presented if the genomic analysis is shown first with the follow-up figures related to the work on auxin and stem cell function. It would also be beneficial if the information on the development of the computational methods for the identification of the SAM center is moved to the supplement with all supporting data and then referenced in the text.

RESPONSE:

All relevant information for our genomic analyses, including replication and pooling, can be found in the methods section. This also extends to a brief comparison to the datasets previously published by us in the main text, as well as the methods section. For example, our published set of direct targets based on ChIP-chip from 2010 included 136 genes at $p < 0.05$, whereas the new ChIP-seq set yielded 6740 regions at the same cut-off.

We disagree with this reviewer concerning the validation experiments. RNA-seq and ChIP-seq with replicates are considered the gold standard in the field and we feel strongly that adding qRT-PCR experiments on a handful of selected targets would not add any additional value. In addition, the PCA plots shown in Fig. 6c show the exquisite reproducibility of our experimental triplicates.

The remaining main text figures two through four and supplemental figures two through five rely largely on the quantification of auxin response as read out by DR5v2 in the shoot apical meristem. The pattern of DR5 response in the shoot apical meristem has been characterized in numerous previous studies. One advance of this study is the analysis of acquisition of auxin response (as referenced by DR5v2) relative to expression of CLV3, which marks pluripotent stem cells in the SAM. The analysis done in figure 1F could be improved if not only the cells of the central zone are shown, but how the transition of high CLV3 expression to high auxin response (DR5v2) occurs spatially. From the images provided in figure D and E it is clear there is overlap in the expression domains for CLV3 and the DR5v2 response. If something could be said on the spatial determination of primordia initiation relative to CLV3 signal as the auxin response increases toward the periphery the paper, and the data of the subsequent figures could be strengthened, it would help to answer the open question of how much of the change in auxin response minimum is due to repression of the auxin signaling response or to expansion of the CLV3 marked stem cells in that central domain shown in figure 1F, in the subsequent data presented in figures two through four.

In figures two through four and supplemental figures 2 through 5 I will outline my comments below as major and minor points.

RESPONSE:

The focus of our work is the modulation of auxin signaling output in stem cells and therefore we feel that the analysis of primordia initiation is beyond the scope of this work.

Major Point 1: The analysis of the data presented herein does not take into account several relevant previously published works. In previously published work either overexpression of CLV3 throughout the SAM or a WUS RNAi throughout the SAM caused meristem termination (1). Either gain of CLV3 function or loss of WUSCHEL function results in no activation of DR5 in the center of the SAM, while termination occurs. In this current study, the authors show that only reducing WUSCHEL protein in the L1/L2, by an anti-GFP nanobody, is sufficient to alter the DR5v2 distribution in the SAM, shifting it toward the center.

In the Yadav paper, an increase in DR5 expression after WUS shutdown is reported, mostly at the periphery.

RESPONSE:

Thank you for bringing up this important point. After comparing a number of different DR5v2 reporters (see also Fig. S3) we are convinced that the discrepancy between the previously published and our data lies in the use of the ER localized form of DR5v2, which allows the identification of much weaker outputs. We have now added a relevant section to the discussion to explain this.

In addition, this current work ignores another figure in the Yadav et al. Development 2010 publication where activation of WUS in the CLV3 expressing cells results in a temporal and spatial expansion in CLV3 expression over time. Importantly, those cells at the periphery of the central zone that had acquired primordium fate, prior to WUS activation, did not express CLV3 during the 120-hour time course. This implies that once primordial fate is acquired those cells are locked into that fate. However, the zone of CLV3 expands radially, around those CLV3 negative cells, while more primordia are formed on the flanks of the enlarged meristem compared to mock treated shoot apical meristems. In light of these results, an explanation for the results shown in Figure 4 panels D and E, where WUS-GR is activated and the expression minimum for DR5v2 is expanded, could simply be due to a cell fate respecification shift toward the CLV3 expression stem cell fate, prior to acquisition of primordial cell fate at the border where cells transiently express both CLV3 and display DR5v2 signal. In fact, evidence for this can be supported by the work of Reddy and Meyerowitz (2), where reduction of CLV3 peptide in the central zone resulted in a respecification of CLV3 non-expressing cells at the border of the central zone back to CLV3 expressing stem cells within a 24 hour window. The authors appear to attempt to address this in supplementary figure 3 and show the CLV3 domain is increased after WUS-GR activation but this is nothing like what is shown for the expansion of CLV3 in other references listed above. If this slight increase is what is observed this would contradict previous studies. For the authors to demonstrate this they would have to show quantification of the CLV3 domain size in great numbers of mock and dexamethasone treated WUS-GR shoot apices. Furthermore, given that the methodologies are different in these referenced studies, the overall experimental approaches should arrive at a similar end (at certain time points), how the authors of this current work reconcile their current results with these published works needs to be addressed.

RESPONSE:

We agree with this reviewer that the issue of cell fate re-specification is a very interesting and exciting issue. However, we do not feel that it is of immediate relevance to our study and we do not make any further statements or claims on this matter. In addition, our results that the

CLV3 domain is only mildly expanded fit very well with the moderate expansion of the DR5v2 negative zone on the centre of the SAM following ectopic WUS expression.

In Bhatia et al. (3), the authors claim that long term induction or expression of a constitutive active form of MP Δ , which lacks domain III and IV, results in transgene silencing. The authors here need to ensure that in the results of Figure 3 and supplementary figure 3 the transgene is active. Also, in Bhatia et al. it is shown that activation of this MP truncated form causes a halt of organ initiation and the authors propose that MP is not permissive but instructive for organ initiation by convergence of PIN1 polarity. If this were the case, I would expect the plants of the pCLV3::MP Δ genotype to show a similar phenotype to that shown in Bhatia et al. where organ initiation is blocked and cells at the apex proliferate into an undifferentiated mass in seedlings where MP Δ is activated. Even though there is a difference in promoters used in these studies, UBQ10 vs CLV3, given that CLV3 is expressed during early embryo development when the apical stem cells are specified, I would assume that expression of MP Δ would give a similar overproliferation phenotype where differentiation on the flanks was terminated. Given the issues of transgene silencing reported by Bhatia et al. the authors need to show evidence that their transgene is expressed at the appropriate developmental time analyzed in figure 3/supplemental figure 2 in the shoot apical meristem. The fact that expression of MP Δ from the At1g76110 HMG promoter results in production of numerous flowers while expression from the CLV3 promoter has little affect the authors need to show that 1.) indeed the stem cell pool is not distributed in the pCLV3:MPD experiment and 2.) The stem cells are maintained in the HMG:MPD experiment while peripheral zone cells are impacted by the presence of MP activity. In addition, the reference given for the MPD is not correct.

The reference given characterizes the genomic loci for MP but does not state anything about the removal of domains III and IV for the generation of MPD. The authors should use Ckurshumova W., Krogan N.T., Marcos D., Caragea A.E., Berleth T. Irrepressible, truncated auxin response factors: natural roles and applications in dissecting auxin gene regulation pathways. *Plant Signal. Behav.* 2012; 7: 1027-1030.

RESPONSE: We agree with this reviewer that it is important to show that our transgenes are active. This is demonstrated using DR5, an in vivo sensor of ARF transcription factor activity. As we show in Fig. 3 j, k, n, DR5v2 is consistently activated in the central zone of plants expressing MP or MP-D from the CLV3 promoter. Still these plants do not develop phenotypes, nor show differences in SAM size (Fig. 3 l).

We thank the reviewer for alerting us to include a specific reference for MP-Delta, which we have now done.

This work also does not reference or address the work of Wu et al. (4), which demonstrates MP works through chromatin remodeling to induce floral primordium fate. Can the authors here provide an explanation of why when a constitutively active form of MP is expressed in the CZ there is not a switch to the PZ cell fate? In this current work WUSCHEL is proposed to work through the HDAC-mediated repression of auxin pathway genes, while the work of Wu et al. would say the SWI/SNF complex would turn on primordium-specific genes. Is this due to WUSCHEL having a dominant role in the stem cells to maintain a repressive chromatin state? A simple test could be done to address if there is a shift in acetylation state of the chromatin in both the pCLV3::MP Δ SAM or the pHMG::MP Δ SAM. Given that primordium production is increased in only the pHMG::MP Δ line you might expect the chromatin isolated from the SAMs of these plants to have a higher percentage of acetylation.

RESPONSE: We agree with this reviewer that this is a fascinating question and have added a sentence in the discussion to explain how our data fit with the Wu et al. results. We feel that addressing this complex issue experimentally would represent another, independent manuscript.

Major point 2: In the use of the pCLV3:iCalSm the mechanism of loss of stem cell identity is not clear. Previous work in reference 5 gave evidence that lack of WUSCHEL movement into these cells could be the mechanism. However, in this figure only the activation of DR5v2 is shown. To make this claim definitively, the authors need to show how the expression of CLV3 changes upon activation of the iCalSm. In reference 5, the time frame presented for meristem termination is 72 hours. This study shows a time course of 120 hours where the meristem appears to still be functioning despite the appearance of DR5v2 in the center of the SAM. It is also not clear why RPS5a is used as the marker for stem cells, when CLV3 expression would be much more appropriate. Yes, RPS5a is expressed in meristematic tissue, but it is also expressed in many other cell types. The authors give no explanation on why this particular promoter is used in lieu of a more traditional stem cell marker.

RESPONSE: Sorry for the confusion. We do not claim that RPS5a is a stem cell marker, in contrast, we use it as a marker of where symplastic isolation has occurred. The RPS5a promoter is active in dividing cells and after plasmodesmata shutdown in stem cells it is turned off. Therefore, the absence of RPS5a activity marks the former stem cells. Using a CLV3 reporter would work the opposite way and would be lost during the experiment. While extremely interesting, we do not feel that the further examination of the iCalSm effects would add much to address how stem cells respond differentially to auxin.

If the closure of plasmodesma in the stem cells does result in their differentiation one point that is not addressed by these experiments is how does the auxin level change given the sharp increase in DR5v2 activity shown. One would expect that the levels of auxin would increase and a change in the R2D2 ratio would be observed. If these data were provided it would further strengthen the authors conclusions give the issues listed above. As stated early, given what is presented in figure 1, it would seem that for all the subsequent experiments the analysis of DR5v2 signal, if done in tandem with CLV3 expression, the concerns outlined above would be answered. Given that in figure 3/supplemental figure 2 the authors ectopically express MP in the CZ or inhibit auxin signaling by the use of a dominant active BDL-D and there no stem cell loss is observed. In addition, when CLV3 is overexpressed in the SAM, where stem cells terminate, KAN1 expression is seen more toward the center of the SAM, indicating the loss of stem cells in the center is coupled to peripheral zone markers moving toward the central region of the SAM (5). This would contradict what is shown in Figure 2 of this current work, where symplastically isolating the CLV3 expressing cells, resulting in their differentiation, does not show any observed disruption in the zonation of the SAM, just appearance of DR5v2 in the center. However, no other specific marker of the SAM stem cell niche is analyzed in these experiments.

RESPONSE: We do agree that these are all interesting issues, but they do go significantly beyond the scope of a single manuscript. We do not address SAM zonation and also do not make any claims about KAN expression.

Major point 3: From the results shown and the discussion at the end of the text, the authors would like to claim that the expression of dominant negative auxin regulators result in SAM arrest, and have used a dominant active form of BODENLOS (BDL-D) to arrive at this claim. However, BDL is not expressed in the center of the SAM but at the periphery according to

previously published in situ hybridization work. This would support their hypothesis. However, in the embryo it is associated with specification of lower basal tier of the embryo, not the apical tier (6). In figure 3, supplementary figure 2, and figure 4 overexpression of the dominant negative form of BDL (BDL-D) from the CLV3 promoter is shown to enlarge the auxin minimum transiently and cause meristem termination if expressed constitutively. However, there is no determination of how BDL-D impacts CZ identity through the analysis of CLV3 expression. The concern is that the expression of BDL-D not just impacts the auxin signaling status but possibly the cell fate when expressed throughout the CLV3 cells, this would be especially relevant in figure 3 and 4 when expressed constitutively from the CLV3 promoter that is activated fairly early in the heart stage. If the authors demonstrated that the CLV3 expression is still present after 24 hours or at some point during the constitutive expression experiments followed by stem cell depletion over time this would strongly support their hypothesis. Furthermore, these results imply that a subset of AUX/IAA genes are repressed in the central zone for the maintenance of the stem cell pool. Given the authors generate data that identifies several AUX/IAA targets as regulated or bound by WUS, which fit the criteria for the mechanism the authors propose, their hypothesis would be greatly strengthened if they could show some of these repressive AUX/IAA genes are turned on, or are upregulated, in the absence of WUS activity, or are repressed when WUSCHEL is expressed in the PZ cells. The only data presented is regarding the expression of either MP and TIR1, where it is not all that clear there is a significant change. To support this is central to their claim that WUS acts to balance auxin signaling in the stem cells of the SAM.

RESPONSE: Thank you for bringing up this important point! We fully agree with the reviewer that the expression of BDL-D could lead to an auxin independent stem cell fate re-specification. To test this directly, we performed a new series of experiments using novel triple transgenic plants that carry pCLV3-BDL-D-GR, pCLV3-mCherry-NLS, as well as DR5v2-er-YFP-HDEL. We analysed these plants by time resolved live cell imaging and can now firmly state that stem cell fate is not directly repressed by BDL-D. In contrast, pCLV3 activity is enhanced following activation of BDL-D and is maintained until termination of the SAM. These important results are now shown in Fig. 3 e-h and supplementary figure 3. We have explained the experiment in the text and refer to it in the discussion.

Major point 4: In reference 5 of the main text, these same authors stated only a C-terminal tagged WUS protein with a 30 amino acid linker can complement the wus null phenotype. Did the authors characterize this N-terminal tagged version for complementation of the wus phenotype? There is no information regarding this point in the text and reading the materials and methods would suggest that this transgene was transformed into wild type Columbia. This is a key point because in their previous report they indicated N-terminal fusion with WUSCHEL lack of complementation implying lack of WUSCHEL function. This line is used for the ChiP-seq, RNA-seq, and imaging experiments of figure 4 and supplementary figure 3.

RESPONSE: The mCherry-GR-linker-WUS fusion protein used here cannot to rescue the wus mutant phenotype, since it is unable to move from cell to cell due to its size. However, similarly designed GFP-linker-WUS fusions do rescue and our mCherry-GR-linker-WUS fusion protein produces many phenotypes after induction that are similar to transcriptionally inducing WUS alone, such as stem twisting and ectopic formation of floral meristems. We have now carried out detailed analyses of the phenotypes caused by inducing mCherry-GR-linker-WUS in the SAM and describe these in the text and the new Supplementary Fig. 5. Importantly, we have never claimed that N-terminal fusions cannot rescue, we actually use one in the paper this reviewer refers to.

Minor point 1: There are no values associated with the y-axis in figure 2G.

RESPONSE: We have corrected this mistake

Minor point 2: In the explanation of the data shown in figure 2 how the callose synthase is expressed is given as reference 5 in the main text. In the materials and methods, it is indicated that the CLV3 promoter was used to drive the expression of callose synthase (iCalSm). I would be informative to those not familiar with this field if an explicit statement is given in the main text or figure legend that the callose synthase was expressed from the stem cell specific CLV3 promoter.

RESPONSE: We have now included this information in the text and figure legend.

Minor point 3: The activation of DR5v2 in the center of the SAM would be much clearer if the data were presented as a Z/Y or Z/X slice in addition to the projection shown. Is this DR5v2 activation in the L1-L3/4, the primary domain of CLV3 expression, or broader in the deeper layers given that the authors have previously shown that cells enlarge after 72 hours in the L3/L4 of the SAM signifying differentiation.

RESPONSE: We have now prepared a new Supplementary Fig. 1 that shows the independent imaging channels for DR5v2, pRPS5a, and cell wall and that includes a Z-projection through the center of the SAM as requested by the reviewer.

Minor point 4: In all figures showing time course data it is not clear in either the main text or in the figure legend if the same shoot apical meristem is being imaged and what was the frequency of imaging.

RESPONSE: We commonly image pools of plants for each time point rather than following a single SAM over time. We have now stated this clearly in the methods section.

Minor point 5: The alleles for *clv3* used in this study are not provided and need to be.

RESPONSE: We used *clv3-10* and this information is now included in the methods section.

Minor point 6: What is meant by nearly all in figure 4C? Please provide numbers to support this claim. Also, the text here indicates seedlings but the image in figure 4 clearly shows a plant with a stem and what appears to be a cauline leaf? This does not look like a seedling. If this is a seedling do they flower early and do the SAMs terminate, after producing few flowers at the seedling stage but still produce a stem that elongates? This would suggest some pool of apical stem cells present to produce the stem tissue?

RESPONSE: We have now repeated the experiment and clearly state in the text that all plants terminate. Most individuals terminate during seedling stage, but some, as the one shown in Fig 4 c make it to the inflorescence stage. We have now spelled this out in the text.

Minor point 7: In figure 4 panels G and H, can the authors address why the WUS-GFP signal is decreased in the basal aspects of the WUS expression domain (toward the pith at the bottom of the rib meristem) in addition to the cell of L1/L2? The expression domain of the CLV3 gene extends at least to the L3 and realistically to the L4. Why is there reduction in GFP signal below the cells of the L3/L4 while maintained in these cells? Is this due to a

feedback mechanism that increases expression from the WUS promoter?

RESPONSE: We have no hard data on this aspect, since we have focused on the effect of removing WUS in the L1 and L2. Having said this, we do see a substantially enhanced activity of the WUS promoter following WUS protein depletion, supporting the hypothesis put forward by the reviewer.

Minor point 8: In the graph shown in Figure 2G there are no values shown on the y-axis. Please provide values and statistical analysis. In the graph shown in figure 4F no X-axis is shown. For clarity it would be good to have an X-axis with a low value starting at 0.

RESPONSE: We have rearranged what is now Fig 2 f and added axis labels, raw data point and statistical analysis.

Minor point 9: The data shown in main text figure 4L for the wus-7 results would be more appropriate if they were moved to supplementary figure 4 where the images for DR5v2 distribution are shown in the wus-7 background. For figure 5 panels B and C, the 24 mock control needs to be shown. If one is to compare the mock treatment in the UBQ10::WUS-GR experiment this really is no change in the distribution of the signal.

RESPONSE: Fig 5 b shows MP expression in a pUBQ10-GFP-NLS plant that expresses the anti-GFP nanobody in stem cells since it has been induced with ethanol. Since the ethanol induction system can cause substantial phenotypes after induction when driven by pCLV3, we very strongly feel that this is a much better control than the mock treatment. Fig 5 d and e demonstrate that WUS activity is able to reduce MP expression in the periphery. We do not claim that the MP expression domain changes. The reduction, but not complete suppression also fits very well with the maintenance of DR5 maxima following ectopic WUS expression shown in Supplementary Fig 6.

Minor point 10: In the main text and materials and methods the WUS-GR transgene is listed as pUBI10, I believe this should be pUBQ10.

Response: We have fixed that gene name acronym.

Minor point 11: The significant digits in Table 2 need to be trimmed.

RESPONSE: We have modified the table accordingly.

Minor point 12: In the materials and methods the concentration for induction of WUS-GR is given but it is not listed for the BDL-D-GR experiments in Figure 3.

RESPONSE: We used 10 μ M for both experiments, this information has now been added to the methods section.

Minor point 13: Please provide the details for the generation of the MP Δ . Does the construct in this work correspond to the same construct reference in major point 1 (3).

RESPONSE: Thanks for bringing up this point. We have now added the relevant information to the methods section.

1. Yadav, R. K., Tavakkoli, M. & Reddy, G. V. WUSCHEL mediates stem cell homeostasis by regulating stem cell number and patterns of cell division and differentiation of stem cell progenitors. *Development* 137, 3581 (2010).
2. Reddy, G. V. & Meyerowitz, E. M. Stem-Cell Homeostasis and Growth Dynamics Can Be Uncoupled in the Arabidopsis Shoot Apex. *Science* 310, 663 (2005).
3. Bhatia, N. et al. Auxin Acts through MONOPTEROS to Regulate Plant Cell Polarity and Pattern Phyllotaxis. *Curr. Biol.* 26, 3202–3208 (2016).
4. Wu, M.-F. et al. Auxin-regulated chromatin switch directs acquisition of flower primordium founder fate. *eLife* 4, e09269 (2015).
5. Yadav, R. K. et al. Plant stem cell maintenance involves direct transcriptional repression of differentiation program. *Mol. Syst. Biol.* 9, 654 (2013).
6. Hamann, T., Mayer, U. & Jurgens, G. The auxin-insensitive bodenlos mutation affects primary root formation and apical-basal patterning in the Arabidopsis embryo. *Development* 126, 1387 (1999).

RESPONSE: We added those of suggested references that were not already cited in our original work.

Reviewers' comments:

Reviewer #1 (Remarks to the Author):

The authors have well revised the manuscript by adding quantification of data and proving proper discussion. The referees' concerns have been generally well addressed.

This paper now nicely supports the idea of stem cells visualized by CLV3 have a differential response to auxin as read out by DR5v2. In addition, they showed that WUS directly binds to many auxin-related genes and represses them through modification of acetylation status of the genome, and they newly provided genomic data mining tool.

Reviewer #3 (Remarks to the Author):

The authors addressed the minor comments sufficiently and for that I thank them. However, I feel they did not address most of the major comments that I had on their work. In fact, in several cases their response was dismissive or handwaving at best. There are several remaining issues that I have that have persisted during the review and revisions. I list them as follows:

1. In this revised draft manuscript there are still numerous instances where figures lack scale bars, graphs lack labeled axis, and the images appear to be displayed to show what the authors want to demonstrate with the 3D projections of the images shown. While I do understand that imaging pools of meristems and different developmental time points, to make the conclusions the authors wish to make it would be advisable to do these induction experiments and follow the changes in the selected reporters in the same meristem samples. These methods are widely published and not outside the scope of a reasonable request. Even if it was just one example from any of their induction experiments.

2. The representation of the data is highly selective and at times confusing given that important controls are missing from the main figure text and are only shown in the supplement (primarily figures 3 and 4 and their corresponding supplemental figures). In addition, for the experiments they wish to draw one of their major conclusions from, the pCLV3::BDL-D and pCLV::BDL experiments, no numbers of plants analyzed are given or the developmental time at which the images represent. They also do not do the proper experiment with the clv3 pCLV::BDL-D control where just the pCLV::BDL is introduced into this background to show that it develops with a clv3 mutant phenotype similar to that of the wild type.

Regarding "We disagree with this reviewer concerning the validation experiments. RNA-seq and ChIP-seq with replicates are considered the gold standard in the field and we feel strongly that adding qRT-PCR experiments on a handful of selected targets would not add any additional value. In addition, the PCA plots shown in Fig. 6c show the exquisite reproducibility of our experimental triplicates." The authors do show in situ hybridizations for TIR1 in the supplement but again these data look similar to the MP in situ hybridizations where it is difficult to assess the extent of repression by WUS or lack of repression in the other genetic backgrounds. The other issue is that these pathway components are the upstream members of the pathway. Their statement here that despite DR5 activation the presence of WUS repression of downstream targets of auxin mediated gene regulation prevents any phenotype is not convincing because no validation is done other than expression from the CLV3 promoter of individual pathway components. A simple experiment of activation of the UBQ10::WUS-GR line or nano-body induction in the inflorescence SAM would go a long way to supporting this statement given they only analyze individual components singularly. To validate this statement the only way to do so would be to analyze the downstream genes of the pathway in gain or loss of function, as they did only for TIR1 or MP. As the manuscript stands now, it is my opinion the data are not substantiated by a valid standardized confirmation experiment (qPCR), confirmation of binding site regulation by reporter gene assay or EMSA, or in situ

hybridization of the downstream genes the authors point to as being repressed by WUS that prevent any visible phenotype. I strongly disagree with their claim a ChIP-seq as being a “gold standard” for this type of work that requires no validation.

3. One final note, the authors also do not take my comment on reordering of the text such that the genomic data (ChIP-seq and chromatin analysis) comes first. If it was ordered in this manner it would make the data presented in figures 1-4 much more understandable and give context to why the induction of the MP or MP-delta does not result in a phenotype. The manuscript should be ordered in a way that aids the reader to understand the hypothesis proposed by the reader. It does not need to be a historical document of the order in which the experiments are done.

We would like to thank both reviewers as well as the editor for their time and input into improving our manuscript. We have now prepared a revised version with changes to text and figures and provide a detailed response to all issues raised below.

Reviewer #1 (Remarks to the Author):

The authors have well revised the manuscript by adding quantification of data and proving proper discussion. The referees' concerns have been generally well addressed.

This paper now nicely supports the idea of stem cells visualized by CLV3 have a differential response to auxin as read out by DR5v2. In addition, they showed that WUS directly binds to many auxin-related genes and represses them through modification of acetylation status of the genome, and they newly provided genomic data mining tool.

Reviewer #3 (Remarks to the Author):

The authors addressed the minor comments sufficiently and for that I thank them. However, I feel they did not address most of the major comments that I had on their work. In fact, in several cases their response was dismissive or handwaving at best. There are several remaining issues that I have that have persisted during the review and revisions. I list them as follows:

1. In this revised draft manuscript there are still numerous instances where figures lack scale bars, graphs lack labeled axis, and the images appear to be displayed to show what the authors want to demonstrate with the 3D projections of the images shown. While I do understand that imaging pools of meristems and different developmental time points, to make the conclusions the authors wish to make it would be advisable to do these induction experiments and follow the changes in the selected reporters in the same meristem samples. These methods are widely published and not outside the scope of a reasonable request. Even if it was just one example from any of their induction experiments.

RESPONSE:

- *We have added scale bars and axis labels throughout.*
- *The images shown in the figures are carefully selected to visualize the trend observed in multiple independent experiments. For each control/experiment combination shown, imaging and analysis settings were identical.*
- *While we agree with this reviewer that following individual SAMs over time allows to address important questions with regards to the behavior and identify of individual cells, it also suffers from severe limitations. The repeated cycles of submersion and imaging impose massive stress to the SAM and therefore the activity of cell signaling reporters invariably is influenced in some of the plants observed. Therefore, it is often difficult to decide what the effect of a perturbation versus the experimental variation is. Since we do not aim to draw conclusions about the signaling state of fate of individual cells in this manuscript, we are convinced that our imaging strategy based on cohorts of plants that are only analyzed once is more robust without sacrificing biological insight.*

2. The representation of the data is highly selective and at times confusing given that important controls are missing from the main figure text and are only shown in the supplement (primarily figures 3 and 4 and their corresponding supplemental figures). In addition, for the experiments they wish to draw one of their major conclusions from, the pCLV3::BDL-D and pCLV::BDL experiments, no numbers of plants analyzed are given or the developmental time at which the images represent. They also do not do the proper experiment with the clv3 pCLV::BDL-D control where just the pCLV::BDL is introduced into this background to show that it develops with a clv3 mutant phenotype similar to that of the wild type.

RESPONSE:

- *We have now moved the controls for Figure 3 e-h from Figure S3 into the main figure and have removed Figure S3.*
- *Numbers for all experiments are of course available in the raw data file and in the cases this reviewer raises, we have analyzed between 50 and 92 independent T1 lines. We have now included these numbers in the main text.*
- *Concerning the new suggestion to use pCLV3::BDL as a control for pCLV3::BDL-D in clv3, there are two important points to take into consideration: 1. Despite the fact that stem cell specific expression of BDL did not cause a phenotype, even the wild-type allele of BDL is a potent, albeit short-lived repressor of auxin signaling output. 2. The effects of stem cell expressed transgenes are often enhanced in the clv3 mutant, likely because of increased pCLV3 promoter activity. One nice example is that the frequency of SAM termination of pCLV3::BDL-D goes from 50% in wild type to 100% in clv3 mutants. Therefore, we believe that this experiment would not represent a good control in this setting, but rather would address the question of BDL dosage required to cause phenotypes. The quantitative ARF/IAA code in stem cells indeed is a highly relevant topic worthy of detailed investigation as laid out in our discussion, but is unfortunately out of the scope of this manuscript.*

Regarding “We disagree with this reviewer concerning the validation experiments. RNA-seq and ChIP-seq with replicates are considered the gold standard in the field and we feel strongly that adding qRT-PCR experiments on a handful of selected targets would not add any additional value. In addition, the PCA plots shown in Fig. 6c show the exquisite reproducibility of our experimental triplicates.” The authors do show in situ hybridizations for TIR1 in the supplement but again these data look similar to the MP in situ hybridizations where it is difficult to assess the extent of repression by WUS or lack of repression in the other genetic backgrounds. The other issue is that these pathway components are the upstream members of the pathway. Their statement here that despite DR5 activation the presence of WUS repression of downstream targets of auxin mediated gene regulation prevents any phenotype is not convincing because no validation is done other than expression from the CLV3 promoter of individual pathway components. A simple experiment of activation of the UBQ10::WUS-GR line or nano-body induction in the inflorescence SAM would go a long way to supporting this statement given they only analyze individual components singularly. To validate this statement the only way to do so would be to analyze the downstream genes of the pathway in gain or loss of function, as they did only for TIR1 or MP. As the manuscript stands now, it is my opinion the data are not substantiated by a valid standardized confirmation experiment (qPCR), confirmation of binding site regulation by reporter gene assay or EMSA, or in situ hybridization of the downstream genes the authors point to as being repressed by WUS that prevent any visible phenotype. I strongly disagree with their claim a ChIP-seq as being a “gold standard” for this type of work that requires no validation.

RESPONSE:

- *We like to point out that we have functionally tested seven genes with roles in auxin signaling (positive and negative regulators), as well 10 genes acting downstream of the signaling cascade, including several members of the TMO, LBD and MYB classes of transcription factors for which over-expression phenotypes have been reported in other tissues. Examples include LBD29, whose over-expression causes autonomous callus formation (Xu et al. 2017), or MYB30, which causes short roots when ubiquitously expressed (Mabuchi et al. 2018), or TMO7 and TMO5 that execute important roles in specification of the hypophysis, or the vasculature, respectively (Schlereth et al. 2010, De Rybel et al. 2013). When expressed from the CLV3 promoter, none of these potent regulators caused phenotypes, supporting the hypothesis that pathway-wide regulation encodes substantial robustness.*
- *To derive target genes, we combine ChIP-seq and RNA-seq after WUS activation, carried out in multiple independent experiments. The only inherent limitation to this large high-quality data-set is the lack of spatial resolution. This can only be afforded by in situ hybridization, which unfortunately suffers from substantial variation and hence makes it a very difficult experimental strategy to validate quantitative differences in mRNA abundance. We have gone to great lengths to show the influence of WUS on the mRNA expression of TIR1 and MP in the SAM and now include new sections in Fig 5. Overall, we now show results obtained from a total of 35 independent SAMs for these two genes using inducible gain and loss of function backgrounds, which we believe is far above the standard in the field. Importantly, we point out the limitations of our dataset in the text and discuss potential future experiments to address this issue.*

3. One final note, the authors also do not take my comment on reordering of the text such that the genomic data (ChIP-seq and chromatin analysis) comes first. If it was ordered in this manner it would make the data presented in figures 1-4 much more understandable and give context to why the induction of the MP or MP-delta does not result in a phenotype. The manuscript should be ordered in a way that aids the reader to understand the hypothesis proposed by the reader. It does not need to be a historical document of the order in which the experiments are done.

RESPONSE:

- *We thank the reviewer again for the encouragement to re-structure the manuscript. However, since we like to maintain the focus of our work on the specific role of auxin signaling in stem cells, rather than on a global analysis of WUS target genes, we prefer to leave the ordering as is.*